# Obesity-associated changes in molecular biology of primary breast cancer

Obesity is associated with an increased risk of developing breast cancer (BC) and worse prognosis in BC patients, yet its impact on BC biology remains understudied in humans. This study investigates how the biology of untreated primary BC differs according to patients' body mass index (BMI) using data from >2,000 patients. We identify several genomic alterations that are differentially prevalent in overweight or obese patients compared to lean patients. We report evidence supporting an ageing accelerating effect of obesity at the genetic level. We show that BMI-associated differences in bulk transcriptomic profile are subtle, while single cell profiling allows detection of more pronounced changes in different cell compartments. These analyses further reveal an elevated and unresolved inflammation of the BC tumor microenvironment associated with obesity, with distinct characteristics contingent on the estrogen receptor status. Collectively, our analyses imply that obesity is associated with an inflammaging-like phenotype. We conclude that patient adiposity may play a significant role in the heterogeneity of BC and should be considered for BC treatment tailoring.

Cancer initiation, development, and progression are largely driven by the interplay between tissues and their microenvironment, which can be heavily reprogrammed when metabolic disorders such as adiposity are present. Adiposity is characterized by excessive, and often abnormal, body fat and generally approximated by the body mass index (BMI). Breast cancer (BC) is one of many types of cancer having been recognized as an obesity-associated disease[1,2]. Obesity (BMI ≥ 30 kg/m²), which has been spreading at a fast pace during the last decades and exerting a negative impact on the health and life quality of women worldwide[3], is an established risk factor of estrogen receptor-positive (ER+) BC in post-menopausal women[4,5] and has also been associated with a higher incidence of triple-negative breast cancer (TNBC)[6,7]. Overweight and obese patients with BC tend to face an increased risk of recurrence and poorer survival as compared to lean patients[8,9]. Additionally, emerging evidence suggests that obesity can result in altered efficacy of systemic therapies[10,11] and increase the complications of local treatments[12,13].

Increasing efforts have been directed to studying the obesity-BC biological link and the most documented mechanisms are often positioned around chronic inflammation, adipokines-related effects, and estrogen and insulin signaling[14]. There is however a significant gap in our current understanding of the connection between adiposity and BC biology in patients, since most of the molecular evidence comes from experimental models[14]. Genomic alterations representing treatment targets or markers of treatment resistance are increasingly used in clinics, such as *PIK3CA, ERBB2,* and *ESR1* mutations, respectively[15–18]. Still, it is not well understood whether the genomic profile of a tumor could differ according to the adiposity of the patient. Interrogation of the correlation between adiposity and the tumor mutational signatures could also shed light on the role of adiposity in carcinogenesis. While the biology of malignant tissues in general has been mostly investigated at the transcriptomic level, only few studies, which were often limited in terms of sample size, have attempted to investigate the adiposity-associated changes in the transcriptome of human breast tumors[19,20]. Furthermore, we need to better understand how adiposity might influence the configuration of the tumor microenvironment (TME) and the interactions occurring between the different cellular compartments. In this study, we sought to exploit large BC data series[21–25] to examine how the genomic and

✉ e-mail: christine.desmedt@kuleuven.be

transcriptomic profiles of treatment-naïve primary BC might differ according to BMI and whether these differences are of potential clinical relevance.

## Results

### Study cohorts

Treatment-naïve primary BC samples from patients with early BC having non-underweight BMI (≥18.5 kg/m²) recorded at the time of diagnosis were identified from the Molecular Taxonomy of Breast Cancer International Consortium (METABRIC)[21], the International Cancer Genome Consortium−BRCA EU project (ICGC)[22], the collection of primary invasive lobular carcinoma samples from European institutions (ELBC)[23], the MINDACT trial[24,26,27], and the BioKey trial[25] (Supplementary Fig. 1). In all cohorts, there were no or modest differences in the tumor characteristics between patients in the investigated subset and all patients in the original series (Supplementary Data 1). Different types of molecular data were available for the studied cohorts (Supplementary Fig. 1). Acknowledging BC molecular and histological heterogeneity, all cohorts were stratified according to histological subtype−invasive carcinoma of no special type (NST) or invasive lobular carcinoma (ILC), as well as the ER and HER2 status (Table 1, Supplementary Fig. 1). Of note, some differences in the tumor characteristics of patients were observed across the cohorts, most probably related to the respective inclusion criteria (Supplementary Data 2).

In subsequent analyses, BMI was considered either as a continuous variable or as a categorical variable of three categories: lean, overweight, and obese. There was no evidence of a difference in the distribution of BMI found between the METABRIC and ICGC cohorts, while the proportion of obese patients was lower in ELBC and MINDACT (Table 1). It was observed in all cohorts that BMI was positively correlated with age and menopausal status, as previously shown[28]. Overweight and obese patients were more likely to be diagnosed with larger tumors and at a more advanced stage in all cohorts (Supplementary Data 3). In MINDACT, the prevalence of NST and hormone receptor-positive (HR+) disease was also higher in obese patients compared to lean and overweight patients (Supplementary Data 3). No statistically evident associations between BMI and other standard clinicopathological characteristics were observed (Supplementary Data 3).

### Association of BMI with driver mutations

A comprehensive list of genes harboring driver genomic alterations in primary BC, including single-base substitutions, small indels, and copy number alterations (CNAs), has been previously reported irrespective of BMI[22] (Supplementary Data 4). Here, we analyzed the differences in the prevalence of these events according to BMI. In the scope of this study, we took into consideration gene-level events for both mutations, which were determined by the presence of mutations classified as oncogenic using a pre-defined classification scheme[29], and CNAs (Supplementary Data 5−6).

We first assessed the association between BMI and BC-specific driver mutations using combined data from the METABRIC and ICGC cohorts for the NST ER+/HER2− and NST ER−/HER2− subgroups, and data from the ELBC cohort for the ILC ER+/HER2− subgroup (Fig. 1, Supplementary Figs. 2−4).

Among patients with NST ER+/HER2−, when considering BMI as a continuous variable, we observed that patients with a higher BMI tended to have higher frequencies of *CDH1* and *TBX3* mutations (Fig. 1a−first column, Supplementary Fig. 2). Considering BMI as a categorical variable, these associations were evident when comparing obese patients to lean patients for *TBX3*, but not *CDH1* (Fig. 1a −third column). On the other hand, *PIK3CA* was less frequently mutated in obese patients compared to lean patients (Fig. 1a). When comparing overweight to lean patients, *PTEN* mutations differed significantly in prevalence (Fig. 1a−second column). In the NST ER−/HER2− subgroup, no statistical evidence for association was found, however, we noticed decreases in the prevalence of *PTEN* and *TP53* mutations in overweight patients as compared to lean patients (Supplementary Fig. 3, 10.6% vs 1.9% and 74.2% vs 57.4%, respectively). Of note, the trend observed for *PTEN* was opposite of that seen in the NST ER+/HER2− subgroup. We observed in the ILC ER+/HER2− tumors several gene mutations with noticeable changes in their prevalence as BMI increased, i.e., increased *ARID1A* and *TBX3* mutations, and decreased *RUNX1* and *TP53* mutations (Fig. 1b−first column). Here, *TBX3* and additionally *PIK3CA* were more and less frequently mutated in obese than in lean patients, respectively (Fig. 1b−third column, Supplementary Fig. 4, 27.3% vs 11.4% and 27.3% vs 43.1%), which was consistent with the observations made for the NST ER+/HER2− subgroup. *TP53* also displayed a similar

## Table 1 | BMI distribution of the investigated subgroups in five patient cohorts

|         |              | Total number of patients | Median (range)        | Lean (%)    | Overweight (%) | Obese (%)  |
|---------|--------------|--------------------------|-----------------------|-------------|----------------|------------|
| METABRIC | NST ER+/HER2− | 215                      | 26.17 (18.55, 46.41)  | 86 (40.0)   | 73 (34.0)      | 56 (26.0)  |
|         | NST ER−/HER2− | 68                       | 25.78 (20.47, 43.46)  | 30 (44.1)   | 24 (35.3)      | 14 (20.6)  |
| ICGC    | NST ER+/HER2− | 177                      | 26.00 (18.70, 51.80)  | 67 (37.9)   | 64 (36.2)      | 46 (26.0)  |
|         | NST ER−/HER2− | 84                       | 26.00 (18.60, 55.40)  | 36 (42.9)   | 30 (35.7)      | 18 (21.4)  |
| ELBC    | ILC ER+/HER2− | 545                      | 23.34 (18.51, 40.86)  | 351 (64.4)  | 143 (25.2)     | 51 (9.4)   |
| MINDACT | NST ER+/HER2− | 735                      | 25.24 (18.59, 68.68)  | 354 (48.2)  | 250 (34.0)     | 131 (17.8) |
|         | NST ER−/HER2− | 118                      | 25.28 (19.00, 39.88)  | 53 (44.9)   | 54 (45.8)      | 11 (9.3)   |
|         | ILC ER+/HER2− | 104                      | 24.19 (19.27, 37.05)  | 65 (62.5)   | 32 (30.8)      | 7 (6.7)    |
| Biokey  | NST ER+/HER2− | 13                       | 24.79 (19.72, 44.08)  | 6 (46.2)    | 3 (23.1)       | 4 (30.8)   |
|         | NST ER−/HER2− | 12                       | 24.13 (22.6, 32.05)   | 8 (66.7)    | 2 (16.7)       | 2 (16.7)   |

Subgroups were determined by histological subtype, and the ER and HER2 status. Only cases with available molecular profiling data, either genomic profiling, bulk, or single-cell transcriptomic profiling, were included.

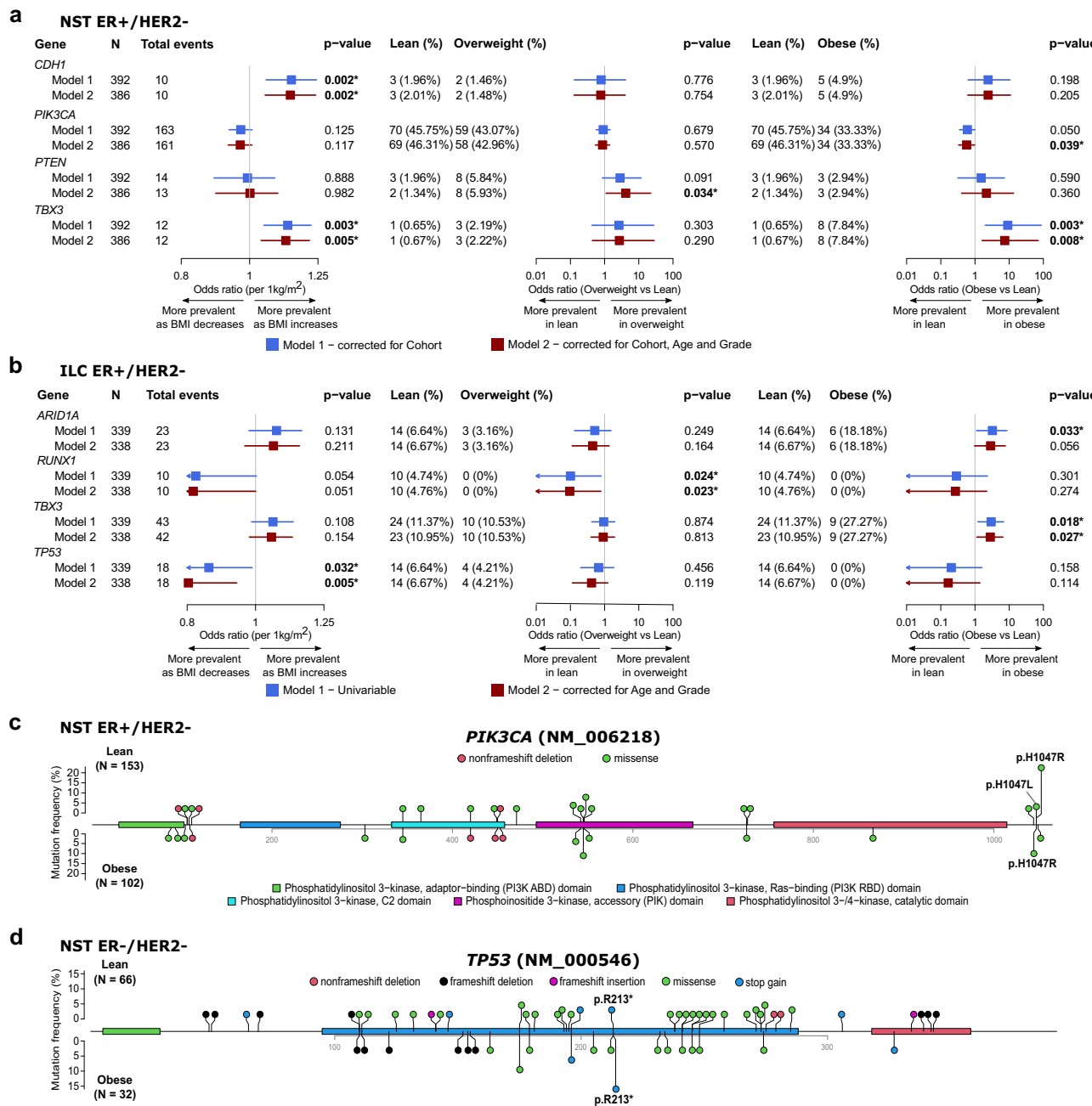

**Fig. 1 | Association of BMI with oncogenic mutations of breast cancer-specific driver genes in patients from the METABRIC, ICGC, and ELBC cohorts.**
**a, b** Forest plots showing the associations evaluated using Firth's logistic regression with *p* value < 0.05 between BMI, either as a continuous variable or a categorical variable (overweight vs lean, and obese vs lean), and driver gene mutations in patients with NST ER+/HER− (**a**) and ILC ER+/HER2− (**b**). Color-coded boxes indicate point estimates of odds ratios, and whiskers indicate their corresponding 95%

confidence intervals. All statistical tests were two-sided. *p* values shown were not corrected for multiple testing. **c, d** Lollipop plots presenting the location and frequency of oncogenic mutations occurring in the *PIK3CA* gene in tumors from lean (top) and obese (bottom) patients in the NST ER+/HER2− subgroup (**c**), the *TP53* gene in the NST ER−/HER2− subgroup (**d**). All statistical tests were two-sided. *p* values shown were derived from Fisher's exact tests and not corrected for multiple testing.

trend to that in the NST ER−/HER2− subgroup, i.e., a lower mutation prevalence in obese and overweight patients (Fig. 1b−third and second column). *RUNX1* mutations were detected exclusively in tumors from lean patients of this ILC subgroup as no event was seen in overweight or obese patients.

Of interest, the association between BMI and the prevalence of *PTEN* mutations in the NST ER+/HER2− subgroup, and *ARID1A* and *ERBB2* mutations in the ILC ER+/HER2− subgroup were better represented by non-linear models (Supplementary Data 7, Supplementary Fig. 5).

We further explored how the distribution of individual oncogenic mutations on driver genes and their prevalence might differ between BMI categories. Two hotspot mutations were found to have a lower and higher prevalence in obese patients than in lean patients with NST ER+/HER2− and NST ER−/HER2−, respectively: *PIK3CA* p.H1047R (Fig. 1c, Supplementary Data 8, 22.2% vs 9.8%, Fisher's exact test *p* value = 0.011) and *TP53* p.R213* (Fig. 1d, Supplementary Data 8, 3.0% vs 15.6%, *p* value = 0.036).

To explore how the tendency of gene mutations to co-occur or be mutually exclusive with each other would change according to

BMI, we performed a Poisson–Binomial distribution-based analysis to identify co-occurring or mutual exclusive pairs of events. In the NST ER+/HER2− subgroup, mutations of the top commonly mutated driver genes in BC, *PIK3CA*, *TP53* and *GATA3*, tended to be mutually exclusive across all BMI categories (Supplementary Fig. 6). However, the mutual exclusivity between the *PIK3CA* mutation and the other two gene mutations was found to be more evident in overweight and obese patients compared to lean patients, which could be linked to the decreased prevalence of this particular mutation in obese patients (Fig. 1a). The mutual exclusivity between *PIK3CA* and *AKT1* mutations, which are usually the activating events of the same pathway, PI3K/AKT/mTOR, was consistently seen in all BMI categories. Our co-occurrence and mutual exclusivity analyses of the NST ER−/HER2− and ILC ER+/HER2− subgroups were constrained by

the lower number of samples, where much fewer gene mutations had a sufficient number of events to be evaluated (Supplementary Fig. 7).

Altogether, our analyses highlight the differences in the somatic mutational profile of patients with BC according to their BMI, which may imply diverse underlying mechanisms contributing to tumor initiation and development.

## Association of BMI with copy number alterations

In a similar manner to the analysis of driver mutations, we examined the association between BMI and recurrent CNAs of BC driver genes and found a number of genes where the prevalence of their amplifications (amp) or hemizygous deletions (hemiLoss) changed according to BMI (Fig. 2, Supplementary Data 9–10).

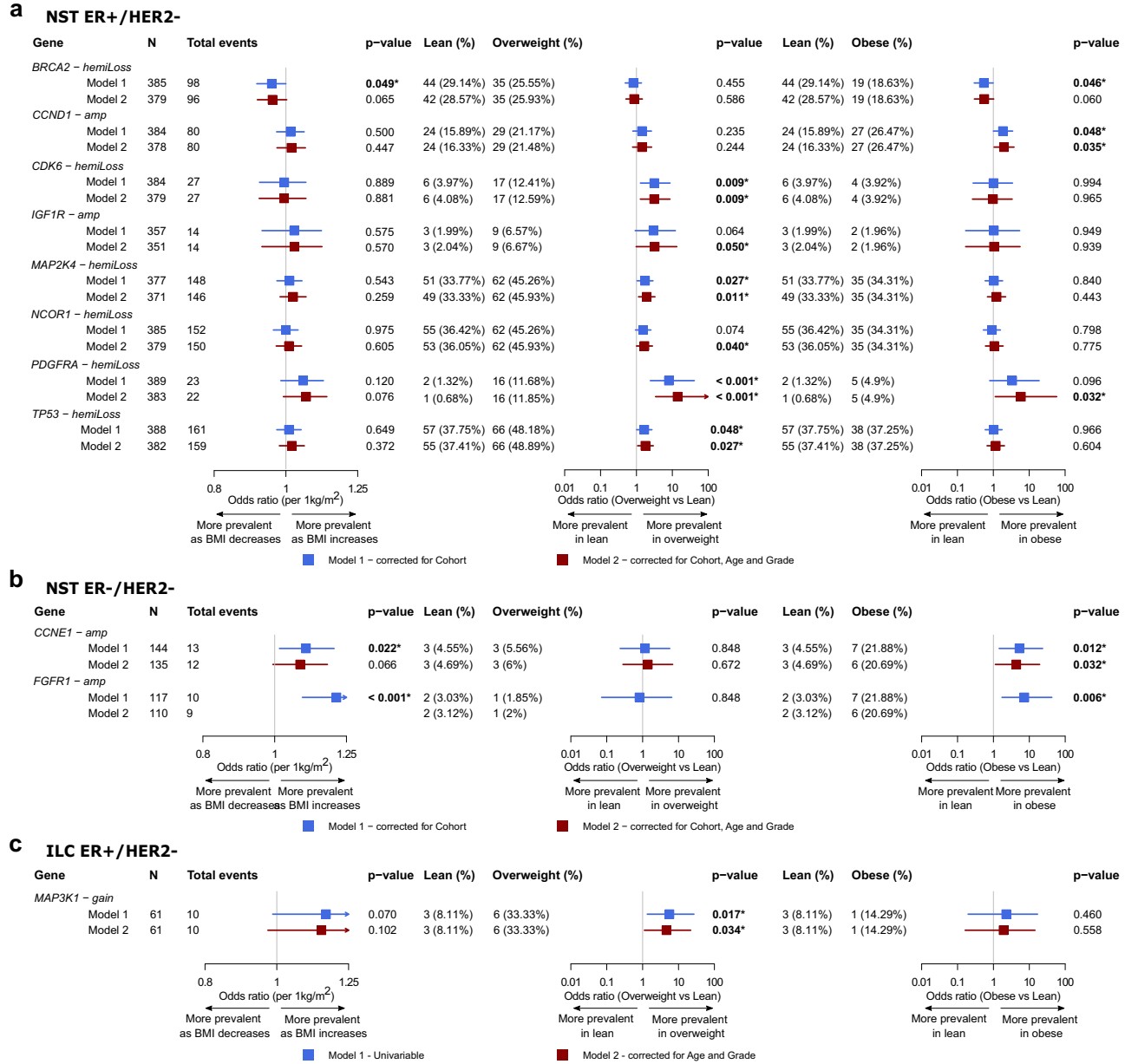

**Fig. 2 | Association of BMI with CNAs of breast cancer-specific driver genes in patients from the METABRIC, ICGC, and ELBC cohorts. a–c** Forest plots showing the associations evaluated using Firth's logistic regression with *p* value < 0.05 between BMI, either as a continuous variable or a categorical variable (overweight vs lean, and obese vs lean), and gene-level CNAs in patients with NST ER+/HER2− (**a**), NST ER−/HER2− (**b**), and ILC ER+/HER2− (**c**). Gene-level CNAs with less than 10 events detected in the respective cohorts were not evaluated. Color-coded boxes indicate point estimates of odds ratios, and whiskers indicate their corresponding 95% confidence intervals. All statistical tests were two-sided. *p* values shown were not corrected for multiple testing. Exact *p* values < 0.001 are specified in Supplementary Data 9–10.

In the NST ER+/HER2− subgroup, the majority of significant associations were positive considering BMI both as a continuous and a categorical variable, meaning BMI-associated CNAs tended to be more prevalent in patients having higher BMI. Among the top 5 recurring gene-level CNAs in patients with NST ER+/HER2− (i.e., *CDH1* hemiLoss, *TP53* hemiLoss, *NCOR1* hemiLoss, *MAP2K4* hemiLoss and *RB1* hemi-Loss), those involving the *TP53*, *NCOR1* and *MAP2K4* genes had elevated frequencies in the overweight category. Less common CNAs such as *CCND1* amp, *CDK6* hemiLoss, *PDGFRA* hemiLoss and *IGF1R* amp were found to be more prevalent in either overweight or obese patients compared to lean patients (Fig. 2a). In contrast, in the NST ER−/HER2− subgroup, *CCNE1* and *FGFR1* amplifications were more frequent in obese than lean patients (Fig. 2b). Using the same criteria for selecting events to be evaluated as the previous two subgroups, we could only analyze a limited number of CNAs for the ILC ER+/HER2− subgroup given the smaller number of samples where this data was available in the ELBC cohort. *MAP3K1* copy gain was the only CNA found to be associated with BMI in this subgroup (Fig. 2c). Non-linear associations between BMI and several CNAs, for instances, *CDK6*, *PDGFRA*, *PTEN* hemiLoss in the NST ER+/HER2− subgroup, and *MAP3K1* copy gain in the ILC ER+/HER2− were suggested (Supplementary Data 9, Supplementary Fig. 8).

With regard to the co-occurrence and mutual exclusivity analyses, we noted evident changes, in the NST ER+/HER2− subgroup, from the lean category to the overweight or obese category in the tendency of co-occurrence between several pairs of clinically relevant gene mutations and CNAs, such as *CCND1* amp/*AKT1* mutation, *ZNF703* amp/*AKT1* mutation, *MDM2* amp/*PTEN* mutation, *NF1* amp/*PIK3CA* mutation, *PTEN* hemiLoss/*PIK3CA* mutation (Supplementary Fig. 6). We found in obese patients the co-occurrence of hemizygous deletion and mutation of the same genes, such as *CDH1* and *TP53*, while not observing the same in lean patients (Supplementary Fig. 6). In the NST ER−/HER2− subgroup, the co-occurrence of *MYC* amplification and *TP53* mutation, which had been reported to be commonly observed in basal-like or triple-negative BC[30], was only statistically evident in obese patients but not in patients of other BMI categories in our data cohort (Supplementary Fig. 7). Assessment of the ILC ER+/HER2− subgroup was hindered by the low number of samples especially those from obese patients and limited CNA calling data.

Here, the findings further support our hypothesis that the landscape of driver genomic alterations of primary BC might differ according to BMI. Consistently observing an increasing trend in the prevalence of numerous putative oncogenic gene-level CNAs in overweight or obese compared to lean patients, we moved forward to inspecting the correlation between genome instability and BMI.

## Association of BMI with genome instability and mutational signatures

To investigate the differences in genome instability, as well as mutational signatures according to BMI, we retrieved relevant data from Nik-Zainal et al. where these genomic features were profiled using whole genome sequencing data of tumors from the ICGC cohort[22].

We first evaluated the association of BMI with genomic instability using the total counts of somatic small mutations, including substitutions and insertions/deletions (indels), and genomic rearrangements as surrogates. In patients with NST ER+/HER2−, the total numbers of somatic substitutions and indels did not appear to differ between BMI categories (Fig. 3a, b, Supplementary Data 11). On the other hand, the count of somatic rearrangements was higher in tumors from overweight compared to lean patients (Fig. 3c, Supplementary Data 11). A slightly higher rearrangement burden was also seen in tumors from obese patients versus lean patients, although with a lack of statistical evidence. No evidence of association between the various measures of genomic instability was found in the NST ER−/HER2− subgroup (Supplementary Data 11).

We next explored the potential association of BMI with changes in the mutational signatures, which revealed remarkable observations in the NST ER+/HER2− tumors. Among the eight single-base substitution signatures and six rearrangement signatures that were evaluated (Supplementary Data 11), a significant increase in the contribution of the substitution signature 1 (COSMIC Mutational Signatures v2, Signature 1) to all single-base substitutions (SBS) was observed in obese patients compared to lean patients (Fig. 3d). Signature 1 has been reported to be correlated with age and its mutational profile represents a mutational process mainly arising from the deamination of 5-methylcytosine at CpG dinucleotides[22,31]. A recent machine learning-based mutational signature analysis showed that while in most cancer tissues, age-associated mutational signatures were represented by elevated contribution of more than one sequence context, in breast invasive carcinoma tissue, the transition S[C > T]G was the sole contributing context of the aging mutational signature[32]. This was reciprocated in our analyses as we observed that changes in the contribution of Signature 1 according to BMI corresponded to a similar pattern in the contribution of its predominant sequence context N[C > T]G (Fig. 3e). Looking further into the subset of somatic SBS detected in BC-specific driver genes that were classified as oncogenic mutations, we found that in obese patients an oncogenic SBS was apparently more likely to be of the sequence context N[C > T]G than in lean patients (Fig. 3f, 9/53 and 2/57, Fisher's exact test *p* value = 0.025). The fact that this mutational signature was associated with BMI independently of age, implied by models adjusted for age and subgroup analyses in different age categories (Fig. 3, Supplementary Fig. 9), suggests that obesity potentially confers similar effects to BC genetics as those by aging.

## Obesity-associated changes in bulk transcriptomic profile of breast cancer

Having identified genomic features associated with BMI, we proceeded to dissect the expression profile of breast tumors to unravel more insights into how their phenotypes might vary according to patients' BMI.

We investigated potential differences in gene expression profile in breast tumors according to BMI categories in MINDACT, the largest cohort with bulk profiling gene expression data available. We identified several differentially expressed genes (DEGs) in tumors from obese versus lean patients with NST ER+/HER2 (Fig. 4a). We then examined the expression levels in different BMI categories for a set of selected genes with known functional roles in the BC-obesity axis[1,2,14] (Supplementary Figs. 10–12). Notable differences in the expression of leptin (*LEP*) and IL-6 (*IL6*) were observed between tumor bulk from obese and lean patients in the NST ER+/HER2− subgroup (Supplementary Fig. 10). DEGs were only identified from the analysis of the NST ER−/HER2− subgroup with a less stringent gene selection (Fig. 4a). In this subtype, the expression of pro-inflammatory cytokines and tissue-repairing factors (*IL6*, *IL1B*, *IL11*, *TNF*, *TGFB1*) was surprisingly lower in tumors from obese patients than from lean patients (Supplementary Fig. 11).

To explore functional changes possibly resulted from indistinct but coordinated changes in the expression of functionally interrelated genes, we performed gene set enrichment analyses (GSEA). Two hallmarks, E2F_TARGETS and G2M_CHECKPOINT, were consistently enriched in tumors from obese patients across all subtypes (Fig. 4b, c, Supplementary Figs. 13–15). These two hallmarks are both involved in cell cycle regulation and their enrichment is usually linked to cell proliferation[33]. c-Myc signaling, one of the key features of TNBC, was further increased with BMI in NST ER−/HER2− tumors. Hallmarks related to inflammatory activities tended to be enriched in tumors from obese patients with either NST or ILC who were ER+/HER2− (Fig. 4b, c, Supplementary Fig. 15). In contrast, these inflammatory hallmarks were enriched in lean patients compared to obese patients

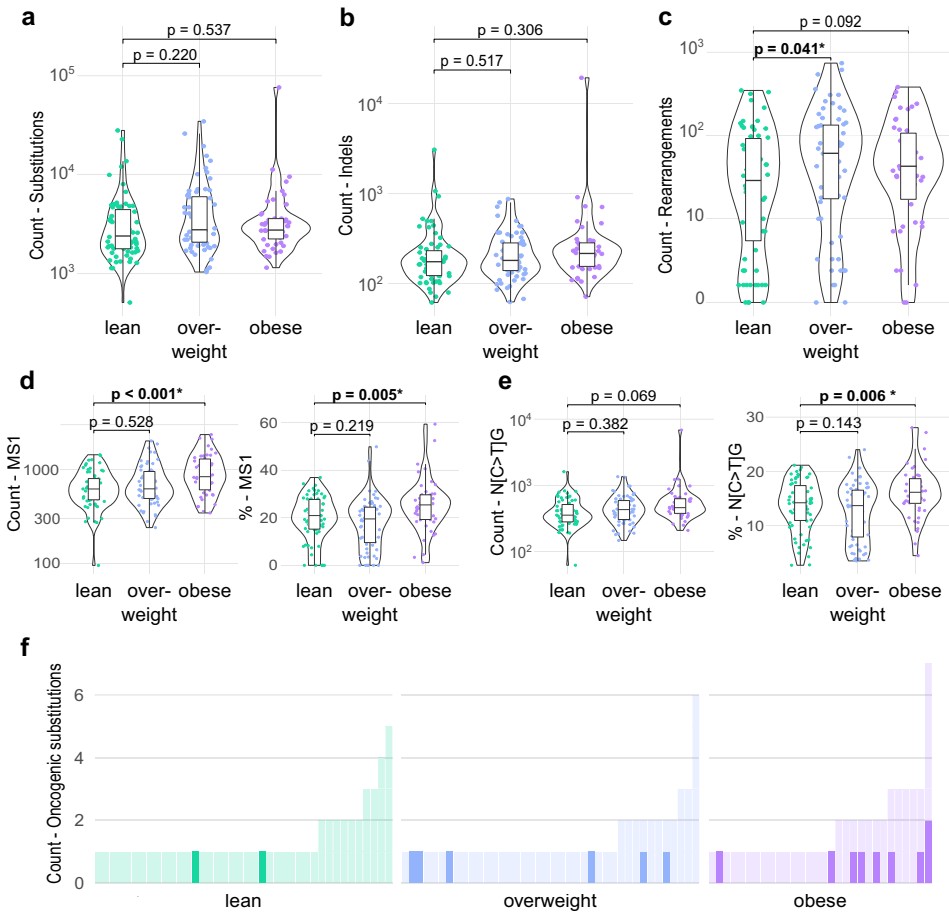

**Fig. 3 | Association of BMI with mutational burden, genomic instability, and the age-associated mutational signature in patients with NST ER+/HER2− from the ICGC cohort.** **a**–**c** Violin/box plots showing the total numbers of substitutions (**a**), small insertions/deletions (indels) (**b**), and rearrangements (**c**) according to BMI categories (lean, $n = 67$; overweight, $n = 64$; obese, $n = 46$). The $y$ axes are log-scaled. **d**, **e** Violin/box plots showing the contribution in absolute count (left) and relative percentage (right) of the mutational signature 1 (MS1) (**d**) and the sequence context N[C > T]G (**e**) according to BMI categories (lean, $n = 67$; overweight, $n = 64$; obese, $n = 46$). In each boxplot, the box denotes the range from the 25th to the 75th percentile, the center line indicates the median value, and the whiskers specify the maxima and minima excluding outliers. All statistical tests were two-sided. Wald test $p$ values determined for coefficients estimated by linear regressions adjusted for age (>50 vs ≤50) and tumor grade (G3 vs G1/G2), are reported and were not corrected for multiple testing. Exact $p$ values < 0.001 are specified in Supplementary Data 11. **f** Bar plots presenting the number of oncogenic mutations occurring in BC-specific driver genes in each tumor. The count of mutations having the sequence context N[C > T]G is highlighted in bold colors. Each bar represents a tumor.

with NST ER−/HER2−, which corresponds to the obesity-associated downregulation of pro-inflammatory cytokines and tissue-repairing factors (Fig. 4b, c). Most of the observations described above for the MINDACT cohort were also seen in the other cohorts with available bulk profiling data, yet disagreeing patterns were observed for some hallmarks (Supplementary Figs. 13–15). Despite the detected associations, BMI as a variable was only able to explain a small fraction of the variation in the tumor biology at the bulk resolution (Fig. 4d).

Comparison of tumor bulk profiles between overweight and lean patients revealed patterns generally resembling those detected in the obese-lean comparison at the hallmark level (Supplementary Figs. 13–18).

Cell fractions were further computationally inferred from bulk expression profiling data of the MINDACT cohort based on a signature matrix of 22 immune cell types[34]. The relative frequency of resting natural killer cells slightly decreased while those of M2-like (anti-inflammatory) macrophages increased in tumors from obese patients, as compared to lean patients of NST ER+/HER2− subtype (Supplementary Figs. 19–21). Resting mast cells, although without statistical evidence, showed a noticeable increase in their relative frequency in tumors from obese patients in both NST subgroups. These results should however be considered with caution given limitations of

current computational deconvolution methods for determining composition of tumor bulk[35,36].

Overall, the bulk profiling was able to depict differences in some of the biological processes in BC tissues between those from obese and lean patients. Since these signals were rather subtle, we hypothesized that obesity has non-homogeneous impact on different cellular populations in the BC TME and therefore postulated that investigation at the single-cell resolution would be a rational direction to proceed.

## Obesity-associated changes in cancer cell-specific transcriptomic profile

We explored the recently published BC-derived single cell BioKey dataset from Bassez et. al. (Table 1, Supplementary Figs. 1, 22), focusing on patients with NST ER+/HER2− (Figs. 5a–h, p, 6a–d) and NST ER−/HER2− (Figs. 5i–p, 6e–h).

We first investigated cancer cell-specific transcriptomic profile and identified more DEGs with more pronounced differences according to BMI than in the bulk profiling (Fig. 5a, i, Supplementary Data 12). Among 17 genes consistently overexpressed in cancer cells from obese versus lean patients in both subgroups, while some of these genes have been reported as markers of proliferation and progression, e.g.,

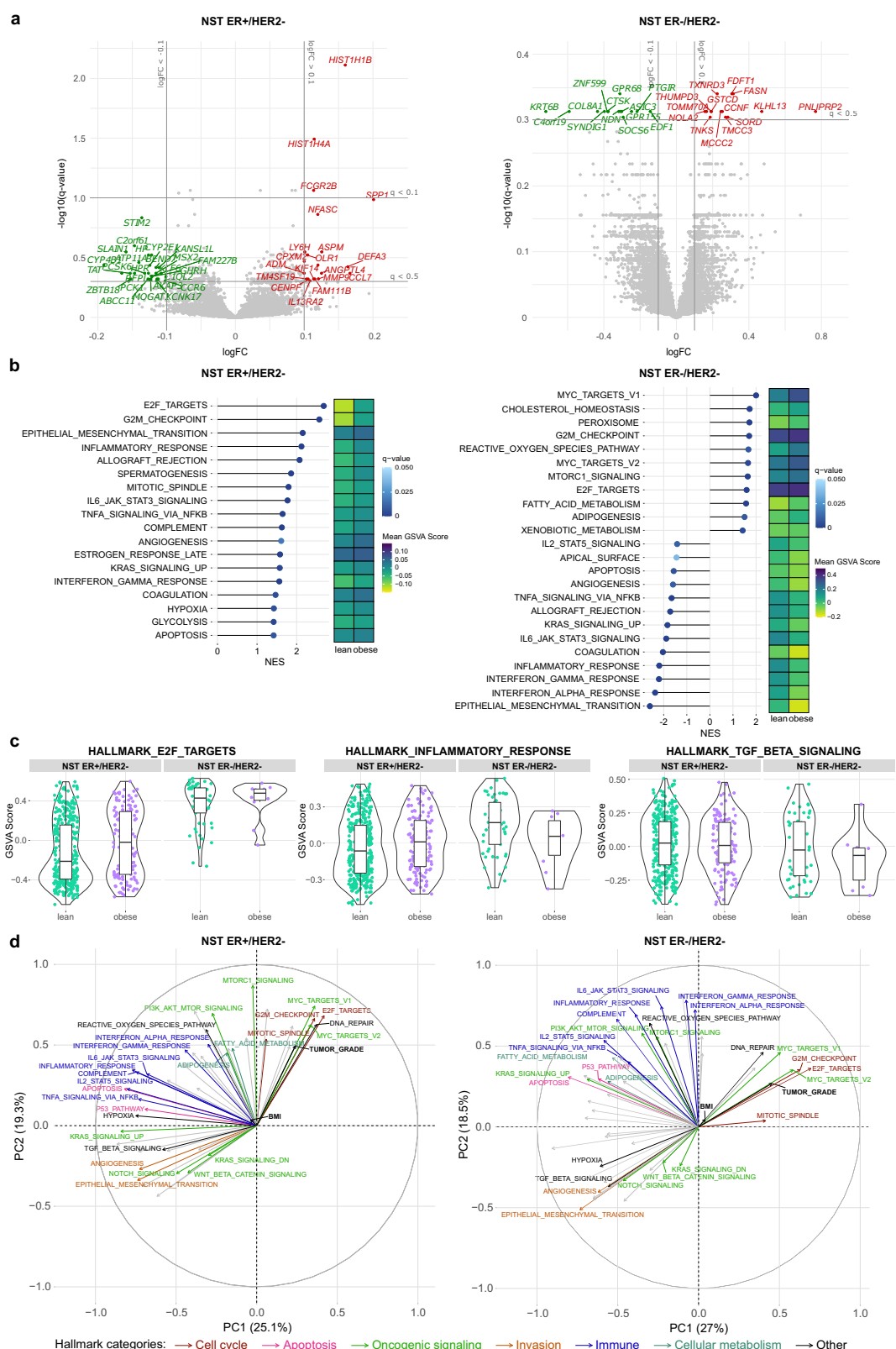

CD24[37,38], claudins (CLDN3, CLDN4)[39], several other genes were thought to be associated with favorable tumor characteristics, e.g., TNFSF10[40], LTF[41] (Fig. 5a, i). Likewise, obesity-associated downregulation of 19 genes was observed in cancer cells of both subtypes (Fig. 5a, i). These include several genes with tumor suppressive roles, e.g., TIMP3[42], CXCL14[43,44]. Exclusively in NST ER+/HER2− tumors, cyclin D1 (CCND1) was elevated, which could possibly be linked to the altered prevalence

of CCND1 amplification according to BMI (Fig. 2a). Other genes that are involved in cell proliferation, migration, invasion, inflammation, and cellular metabolism, and might be relevant for further investigation, e.g., mucins (MUC1, MUCL1, MUC5B)[45,46], inflammatory signaling factors (FOS, JUNB, IL32)[47,48], insulin receptor INSR, lipid transporter APOD[49], were also found to be overexpressed in NST ER+/HER2− cancer cells from obese patients (Fig. 5a). Notably, NST ER−/HER2− cancer

**Fig. 4 | Obesity-associated differences in the transcriptomic profile of primary breast cancer detected from the bulk profiling of tumors from the MINDACT cohort. a** Volcano plots showing differentially expressed genes (DEGs) comparing tumors from obese patients and those from lean patients. Gene expression data are presented as $\log_{10}$-ratio expression values. Genes with absolute log-fold change (logFC) > 0.1 and $q$ value < 0.1 are colored and labeled (red: upregulated in tumors from obese patients, green: upregulated in tumors from lean patients). **b** Lollipop plots displaying differentially enriched molecular hallmarks according to BMI category (obese vs lean) detected by GSEA ($q$ value < 0.1) and heatmap showing their corresponding average enrichment scores computed by gene set variation analysis (GSVA). The lengths of the lollipops represent the absolute values of the normalized enrichment scores (NES). The signs of the NES indicate the orientation of the differential enrichments (positive: enriched in tumors from obese patients, negative: enriched in tumors from lean patients). **c** Violin/box plots of GSVA scores of a cell cycle-related (E2F_TARGETS), an immune-related (INFLAMMATORY_R-ESPONSE), and a wound healing-related (TGF_BETA_SIGNALING) hallmark in NST

ER+/HER2− (lean, $n = 354$; obese, $n = 131$) and NST ER−/HER2 tumors (lean, $n = 53$; obese, $n = 11$) from obese and lean patients. In each boxplot, the box denotes the range from the 25th to the 75th percentile, the center line indicates the median value, and the whiskers specify the maxima and minima excluding outliers. **d** Loadings of the fifty hallmarks, continuous BMI, and tumor grade on the first two principal components (PCs). The principal component analysis (PCA) was performed on a matrix consisting of rows representing patients in the BMI categories lean and obese, and columns representing the GSVA scores of the fifty hallmarks. The coordinates of the two clinical variables were then predicted based on the determined PCs. The angles between the vectors are informative of how they correlate with one another, and the lengths suggest the influence of the variables on this specific two-dimensional space. Hallmarks of importance in the context of cancer are labeled and colored according to their functional categories. Percentage of explained variability by each PC is reported in the axis label of the corresponding axis.

cells from obese patients expressed lower levels of major histocompatibility complexes class I (MHC-I) (*HLA-B*, *HLA-C*) (Fig. 5i), suggesting a potential niche for evasion of anti-tumor immunity[50]. Differential gene expression analyses (DGEA) of cancer cells from overweight versus lean patients of both subtypes also revealed differences in their expression profiles (Supplementary Fig. 23a, b, Supplementary Data 13), however with marginal overlaps with the obese versus lean analyses. This could mean diverse association of BMI along its spectrum to the expression profile of the cancer cell population, but could not yet be verified due to limited numbers of patients. Nevertheless, these detected changes hint at a possible reprogramming of mammary epithelial cells in an obese setting via a complex and varying combination of cellular and metabolic processes.

## Obesity-associated changes in non-malignant cell type-specific transcriptomic profile and the TME

Inspection of cell type-specific differential expression in non-malignant cells using the BioKey data revealed an elevated inflammation in the obesity context. In NST ER+/HER2− tumors, this inflammation showed signs of multi-directionality, owing to simultaneous differential enrichment of contradictory pathways in various cellular compartments, e.g., (I) overexpression of antigen-presenting genes in B cells and mast cells, (II) downregulation of interferon (IFN) response genes in T cells; (III) overexpression of pro-inflammatory and wound healing-like pathway genes in fibroblasts; and (IV) overexpression of pro-inflammatory genes as well as anti-inflammatory genes in endothelial cells (Fig. 5b–h, p, Supplementary Figs. 24–25, Supplementary Data 12, 14–15). In NST ER−/HER2− tumors, there were also hints of a multi-directional and unresolved inflammation in tumors from obese patients but with different molecular characteristics from those in the NST ER+/HER2− subtype, e.g., (I) downregulation of antigen-presenting genes in B cells and dendritic cells, (II) overexpression of IFN response genes in T cells and macrophages/monocytes (Mf/Mono), (III) overexpression of IFN response, invasion-supportive, and wound-healing like genes in fibroblasts; and (IV) IFN response, pro-inflammatory genes in endothelial cells (Fig. 5j–p, Supplementary Figs. 26–27, Supplementary Data 12, 14–15). Notably, we detected in T cells from NST ER−/HER2− obese patients increased expression of immune checkpoint genes (*PDCD1*, *TIGIT*) (Fig. 5o, p). Of note, the expression profile of mast cells in NST ER−/HER2− tumors could not be evaluated due to a low absolute number of cells captured in the data (Supplementary Data 18). Most of these patterns were also observed in tumors from overweight patients versus lean patients of both subtypes, albeit with more subtle signals (Supplementary Fig. 23c, Supplementary Data 13, 16–17). These observations suggest that the TME in the obesity context might be associated with a complex inflammatory profile without a clear orientation, suggestive of a TME with

unresolved inflammation. However, the nature of this inflammation was not identical in the two subtypes, and with the one in NST ER−/HER2− additionally displaying wound healing-like elements.

To confirm the presence of such a TME in obese patients, we first looked at the TME composition, followed by an analysis of single cell-cell communication to dissect the inflammatory signaling characteristics of obese and lean patients[51]. In NST ER+/HER2− tumors from lean patients, fibroblasts were the most abundant non-malignant cell type followed by T cells, while in those from obese patients, T cells occupied the predominant quantitative position, followed by fibroblasts (Fig. 6a, Supplementary Data 19). Mast cells were seen to be more frequently present in NST ER+/HER2− tumors from obese patients (Fig. 6a, Supplementary Data 19), which agrees with the observation from the deconvolution analysis of the bulk data. There was also a shift in cellular proportions in the TME of NST ER−/HER2− tumors between obese and lean patients, with fibroblasts increasing and macrophages/monocytes decreasing, while T cells remained the most prevalent cell type (Fig. 6e, Supplementary Data 19). In terms of intercellular signaling, comparable numbers and strength of putative interactions were computationally estimated for NST ER+/HER2− tumors from lean and obese patients (Supplementary Fig. 28a). In tumors of this subtype from both lean and obese patients, fibroblasts and endothelial cells were responsible for the bulk of intercellular interactions with T cells, macrophages/monocytes and mast cells showing considerable variability depending on lean vs. obese status (Fig. 6b). Accordingly, signaling interactions between fibroblasts or endothelial cells vs. cancer cells, and between mast cells vs. all other cell types in the TME, increased prominently in tumors from obese, as compared to tumors from lean patients (Fig. 6b, c). A tissue-level inflamed state was suggested, substantiated by specific pathways overrepresented in obese patients, such as CCL, B cell regulatory CD22, CD45 (Fig. 6d). Hints of multi-directional immunoregulatory activities were present and characterized by the enrichment of pathways such as SEMA4, SEMA3, FGF (Fig. 6d). Assessing the TME of NST ER+/HER2− overweight patients, we also generally observed an increase in inflammatory response-related interactions, for instance, B cell and mast cell signaling (Supplementary Fig. 29a–d). In the NST ER−/HER2− subgroup, tumors from obese patients appeared to be more active in terms of cell-cell crosstalk with more and stronger interactions (Supplementary Fig. 28b). There was more differentiation in the roles of different cell types in these tumors, with fibroblasts and endothelial cells emerging as the two main sources of signaling (Fig. 6f). Here, the fibroblast-endothelial network remained the driver of the obesity-associated changes in the TME intercellular communication, except that the crosstalk occurred largely amongst themselves, instead of also involving cancer cells or mast cells as in the NST ER+/HER2− tumors (Fig. 6f–g). Inflammation in tumors from obese patients was also elevated, however its characteristics here were much more prominently

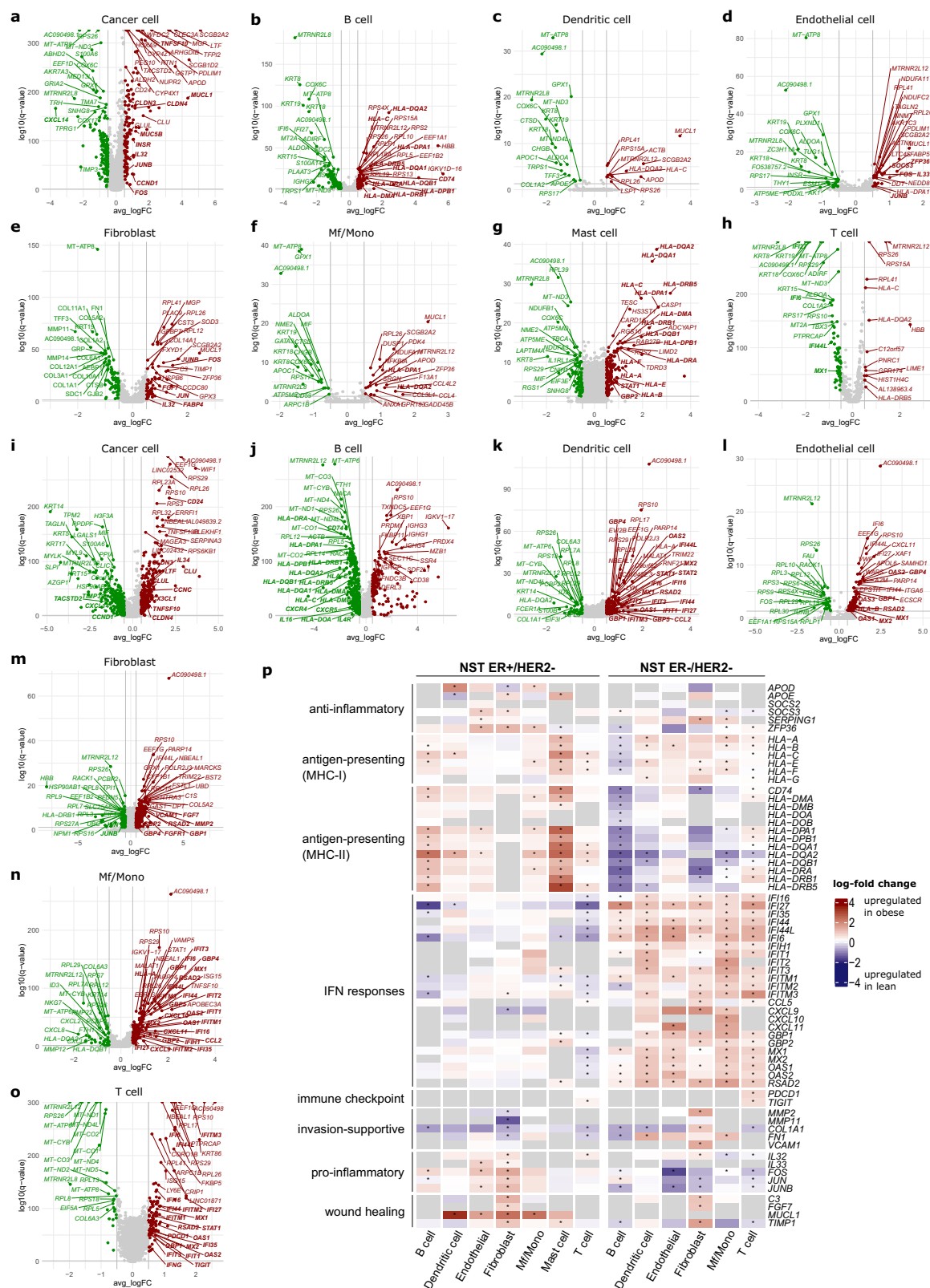

oriented toward wound healing-like or tissue repair-like signaling, represented by pathways led by PERIOSTIN, VCAM, FGF, CSF, PTN, PDGF, TENASCIN, SEMA6, NOTCH, PDL2 (Fig. 6h). The TME of overweight patients of this subtype strongly resembled that of obese patients in terms of its wound healing-related elements (Supplementary Fig. 29e–h).

Taken together, this emphasized that obese patients possess a more chronically inflamed TME. However, depending on the BC-subtype, there were prominent differences in the molecular characteristics of these pathways thereby emphasizing a complex interplay of convergent and divergent inflammatory pathways behind BC-obesity crosstalk.

**Fig. 5 | Obesity-associated differences in the cell type-specific transcriptomic profile of primary breast cancer detected from the single-cell profiling of NST ER+/HER2− and NST ER−/HER2− tumors from the BioKey cohort. a–h** Volcano plots highlighting cell type-specific differentially expressed genes (DEGs) in eight cell types between obese and lean patients in the NST ER+/HER2− subgroup. Genes with absolute log₂-fold change (logFC) > 0.5 and q value < 0.05 are colored (red: upregulated in cancer cells from obese patients, green: upregulated in cancer cells from lean patients). The top 20 upregulated (sorted by q value), 20 downregulated genes, and genes discussed in the main text (in bold) are labeled. **i–o** Companion

plots of (**a**–**h**) for the NST ER−/HER2− subtype. Mast cells were excluded from the analyses for this subtype due to low absolute cell counts. **p** Heatmaps showing differential expression of a selection of genes involved in several immune and cancer pathways in non-malignant cell populations. The cell color is scaled based on the log-FC values (obese vs lean) estimated by the MAST test. Gray cells indicate genes not being tested due to expression in less than 10% of the corresponding cell type in both BMI categories. p values shown were adjusted for multiple testing using the Benjamin−Hochberg method (presented as q values). *, q value < 0.05. MHC-I, major histocompatibility complex class I; MHC-II, MHC class II.

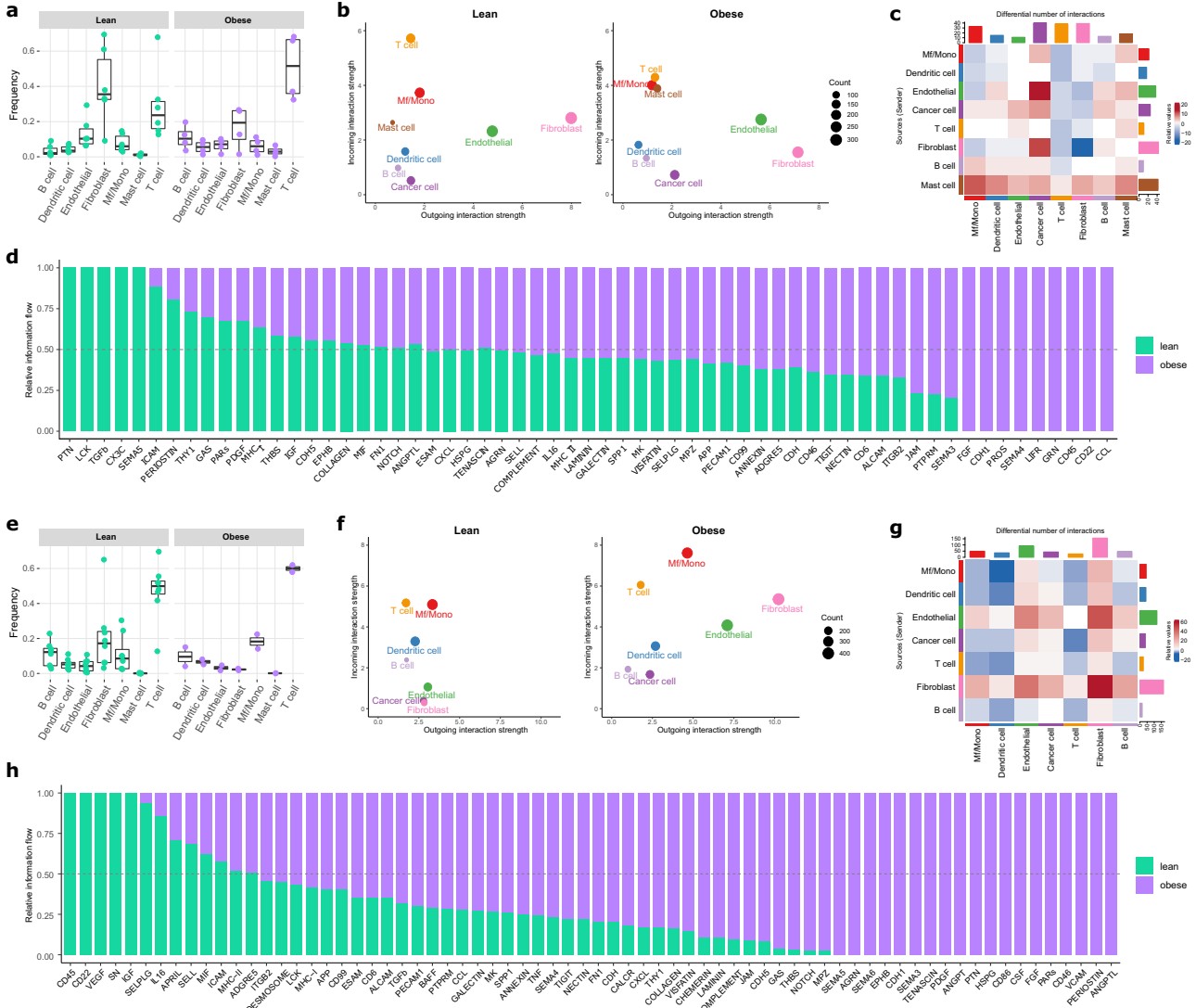

**Fig. 6 | Obesity-associated differences in the cell composition and intercellular interactions within the tumor microenvironment of primary breast cancer detected from the single-cell profiling of NST ER+/HER2- and NST ER−/HER2- tumors from the BioKey cohort. a, e** Frequencies of non-malignant cell types relative to the non-malignant cell pool for NST ER+/HER2- (**a**) and NST ER−/HER2- (**e**) lean and obese patients (NST ER+/HER2− lean, n = 6; obese, n = 4; NST ER−/HER2− lean, n = 8; obese, n = 2). Each data point represents the frequency of the corresponding cell type detected in an individual sample. In each boxplot, the box denotes the range from the 25th to the 75th percentile, the center line indicates the median value, and the whiskers specify the maxima and minima excluding outliers.

**b, f** CellChat-derived outgoing and incoming interaction strength from and to cancer cells and non-malignant cells in NST ER+/HER2- (**b**) and NST ER−/HER2- (**f**) tumors from lean and obese patients. **c, g** Differential number of interactions in NST ER+/HER2- (**c**) and NST ER−/HER2- (**g**) tumors detected by CellChat between lean and obese. **d, h** Relative Information flow of signaling pathways in the intercellular communication network in NST ER+/HER2- (**d**) and NST ER−/HER2- (**h**) tumors from lean and obese patients. The relative information flow was estimated by the sum of CellChat-derived communication probability between all pairs of cell compartments in the network. Signaling pathways with non-zero information in at least one of the BMI categories are shown.

## Discussion

So far, the association between the molecular features of BC and patient adiposity remains largely unexplored in humans. As an effort to reduce the knowledge gap, we retrospectively analyzed data from several large-scale BC studies, which constitute the largest patient series with available BMI to date, and revealed molecular features associated with BMI, some of which with potential clinical relevance (Fig. 7).

Clinical utility of genomic alterations has been proven an advantageous approach to precision oncology, with many clinically

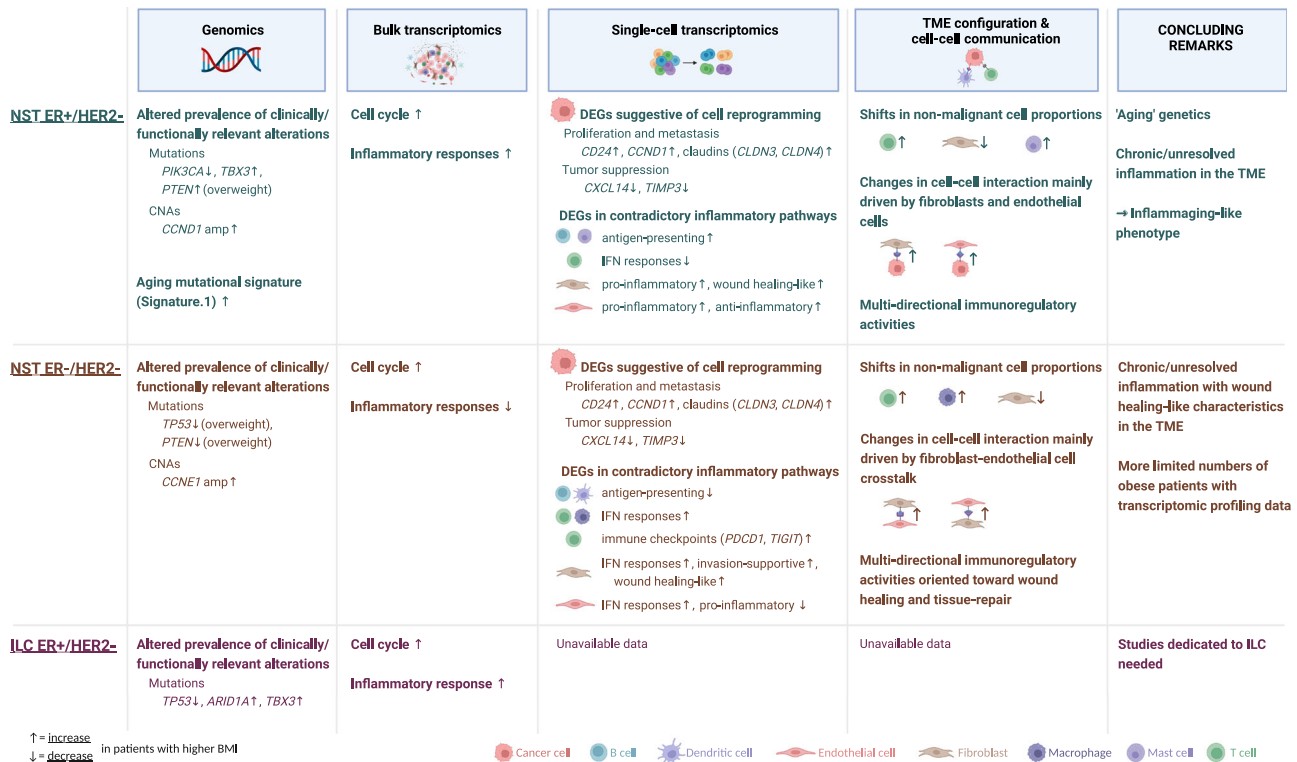

**Fig. 7 | Overview of the main findings from the analyses of different datasets for the three subgroups NST ER+/HER2−, NST ER−/HER2−, and ILC ER+/HER2−.** Observations involving features of known or potential clinical relevance and those hypothesized to imply biological heterogeneity are highlighted. Up and down arrows accompanying highlighted features denote, respectively, increase and decrease in either prevalence, expression level, or frequency in obese, or overweight if specified, compared to lean patients. Created with BioRender.com.

actionable alterations having been established and recognized[15,52]. Here, we demonstrated that the landscape of somatic driver genomic alterations in breast tumors differs according to patients' BMI at diagnosis. Mutation of *PIK3CA*, usually an indicator of induced PI3K/AKT/mTOR signaling and a marker predictive of response to the PI3K inhibitors in hormone receptor (HR)-positive BC patients[15,16], was found to occur less frequently in obese patients with NST ER+/HER2−. In the condition of excess adiposity, PI3K signaling pathway can also be over-stimulated in the absence of an activating *PIK3CA* mutation as a result of multiple changes in the activities of its regulators, such as leptin upregulation and adiponectin downregulation, increased insulin/IGF signaling and overexpression of proinflammatory factors IL-6 or TNF-α[1,53,54]. As cells are able to proliferate through more mechanisms, the pressure selection for tumor cells harboring an activating *PIK3CA* mutation, for instance H1047R as seen in our data, would possibly be lower in the obese setting. This could potentially render this gene mutation less informative to select obese patients for PI3K targeting therapies. Other somatic alterations having been presented with evidence as potential predictive markers for various therapeutic approaches, for examples, *CCND1* and *CCNE1* amplifications for CDK4/6 targeting therapy in ER+ and TNBC, respectively[55,56], were found more frequently in obese patients. Therefore, it could be worthwhile taking adiposity status into consideration for evaluation of the predictive value of these markers in clinical trials. Furthermore, future studies of mechanisms underlying different selection of oncogenic genomic alterations according to adiposity status are warranted for better comprehension of these associations. Novel findings regarding altered prevalence in ER+/HER2− tumors according to BMI of somatic mutation of the *TBX3* gene, which is involved in a complicated and extensive gene regulatory network[57], were made but require further investigation to infer their implications, especially in the cancer-obesity cascade.

Obesity has been widely considered an age accelerating factor demonstrated by many of its biological characteristics shared with aging[58–60]. Here, our analyses showed part of the connection between obesity and aging through a common mutational process represented by the mutational Signature 1. Our observations pointed to their similar effects on the genetics of breast tumors, particularly in NST ER+/HER2−. Our current results also further supported the hypothesis suggested by Afsari et al. that one of the ways obesity promotes carcinogenesis is by giving rise to a specific mutagenesis rather than by accumulation of somatic mutations in cells[32]. As age and obesity are both established risk factors for BC, additional evidence of their interconnection with each other and with cancer further reinforces the importance of tackling obesity to alleviate its risk effects either on its own or in combination with age.

Exploring the gene expression profile of breast tumor tissue, first at the bulk resolution, we observed several obesity-associated differences that were consistent with earlier data of the biological relationship between BC and obesity. These included aberrant cell cycle regulation in all subtypes, and increased inflammatory responses in the ER+/HER2− tumors[61–64]. These differences were however subtle and there was a lack of correspondence in our findings at the bulk mRNA level with established functional differences in BC according to adiposity status, which were mostly shown on the protein level in preclinical models[61,65–68]. Hence, we speculate that data generated from bulk samples might not be robust to investigate the transcriptomic profile of the tumor microenvironment which could potentially be highly cell type-specific.

Our exploration at the single-cell resolution revealed preliminary yet intriguing insights that could be of high relevance for further investigation and confirmed that the single-cell approach was indeed a promising strategy to complement the traditional bulk-level analysis. Cancer cells from obese and lean patients

generally showed measurable differences in their gene expression profiles. Given the broad functional landscape of epithelial cells and their heterogeneity, it was still challenging to precisely infer the functional implication of these transcriptomic-level differences. However, observations implying obesity-driven changes in the expression profile of the cancer cell population, which might lead to changes in the behavior of the disease in biological, prognostic and therapeutic contexts, were made and could be further investigated and validated. Remarkably, in contrast to our knowledge where there has not been any report highlighting mechanistic discrepancy in obesity-induced immune response in different BC molecular subtypes, our data suggested that potential impact of obesity on the immune landscape of the BC TME might differ according to the ER status. It was observed that in both ER+ and ER− tumors, obesity promoted chronic or multi-directional inflammation, which is suggestive of a pro-tumorigenic niche[69,70]. However, the nature of these changes was different according to the ER-status based on our current data, which might hypothetically have major repercussions for treatment strategy. Further validation and mechanistic investigation in this direction could pave the way for designing therapeutic combinatorial treatments against BC in the obesity context. Importantly, the observational findings of this study need to be extended with analyses of healthy controls.

Our study has several existing limitations. Firstly, as BMI was retrospectively collected for most of the cohorts in this study, it was not available for a significant part of the original series. Secondly, there existed differences in clinical and pathological characteristics of patients in different cohorts, making comparison and validation of analysis results across data sets not straightforward. Thirdly, interactions of BMI with other clinicopathological features could not be completely assessed, especially for small cohorts such as Biokey. Nevertheless, we adjusted, where possible, for important features with prominent impact on the tumor molecular biology such as age, menopausal status and tumor grade in our analyses. Finally, although BMI is a conveniently accessible metric, it may not always be an accurate indicator of metabolic health related to adiposity[71]. We intend to address these limitations and further extend the preliminary findings of this study in a prospective study where we will be investigating the TME according to adiposity at the single-cell resolution in a larger series and exploring other anthropometric and histopathological measures of adiposity in addition to BMI (https://clinicaltrials.gov/ct2/show/NCT04200768). In-depth characterization of all cell populations present in the BC tissue, including adipocytes, more scarce immune cell types such as mast cells and dendritic cells, as well as their phenotypes, will be performed in this study. Tumor-adjacent normal tissues will also be available and analyzed within the scope of this study.

In conclusion, we present in this work molecular features of primary BC that differ according to patients' BMI. A number of genomic alterations used or studied as biomarkers in BC, which had altered prevalence in tumors from overweight and obese patients, were revealed. We further emphasize the importance of tackling obesity in BC management and prevention by reporting additional evidence of the obesity-aging-BC interconnection. We uncovered aggregated evidence from analyses of both genomic and transcriptomic data that obesity promotes an inflammaging phenotype of BC[72]. We also highlighted that obesity might have diverse impact to the BC immune landscape according to the ER status of the tumor, a finding requiring more extensive investigation due to its potential influence on treatment approaches, particularly immunotherapy. This study is one of the first to explore the single-cell approach for studying the interplay between obesity and BC and was able to demonstrate it is indeed an advantageous strategy to be used in future research.

## Methods

### Patients and data collection
We requested access to or retrieved from either original publications or open data portals clinical data and molecular data of primary tumors from five BC patient cohorts: METABRIC from Curtis et al. and cBioPortal, ICGC from Nik-Zainal et al. and ICGC Data Portal (DCC Release 28), MINDACT from Jacob et al., ELBC from Desmedt et al., and BioKey from Bassez, Vos et al.

BMI was represented both as a continuous variable and as a categorical variable of three categories according to the World Health Organization (WHO) criteria: lean ($18.5 \leq BMI < 25 \, kg/m^2$), overweight ($25 \leq BMI < 30 \, kg/m^2$) and obese ($BMI \geq 30 \, kg/m^2$).

We were able to retrieve genomic alteration data derived from bulk DNA sequencing and genome-wide SNP array for METABRIC, ICGC and ELBC, bulk gene expression data generated by DNA microarray for METABRIC (Illumina), ELBC (Affymetrix), MINDACT (Agilent), and by RNA-seq for ICGC, and single-cell gene expression data generated by single-cell RNA-seq for BioKey (Supplementary Fig. 1). Patients were stratified according to the histological classification and the status of ER and HER2 of their primary tumors. Due to a small number of available cases, patients with HER2+ tumors were excluded from our current study. Subsequent analyses focused on the three main subgroups of patients: NST ER+/HER2−, NST ER−/HER2−, and ILC ER+/HER2−. To increase the sample size, we combined data of somatic genomic alterations from the METABRIC and ICGC for two subgroups NST ER+/HER2− and NST ER−/HER2−. The ILC patients from these cohorts were excluded provided the small numbers. Further details of the data flow from collection to patient selection and patient stratification, as well as the number of samples with available data for each type of molecular data, can be found in Supplementary Fig. 1.

### Classification of somatic mutation calls and determination of gene mutation
Mutations, including substitution and small indels, were classified as one of the following categories according to the corresponding definition described by Desmedt et al.[29]: Oncogenic, Putative oncogenic, Possible oncogenic, and Unknown significance. We selected only oncogenic, putative, and possible oncogenic mutations for determination of gene mutation status and they were all referred to as 'oncogenic' in the text for simplicity. A gene mutation was determined to be present if there is at least one oncogenic mutation detected in the gene, and absent otherwise. Gene mutations to be evaluated in downstream analyses were limited to genes that were previously reported to harbor driver mutations in primary BC by Nik-Zainal et al. (Supplementary Data 4).

### Identification of gene-level CNAs and oncogenic CNAs
Data of gene-level somatic CNAs for the METABRIC cohort were available for download in the cBioPortal repository (08 April 2019). Somatic CNA events in this dataset were distinguished between four categories, homozygous deletion, hemizygous deletion, low-level gain, and high-level amplification. Copy number segmentation calls of the ICGC cohort were classified as homozygous deletions and amplifications using the definition described by Nik-Zainal et al. (homozygous deletion: copy number = 0; amplifications: copy number ≥5 with ploidy <2.7n, or copy number ≥5 with ploidy >2.7n). The remaining copy number losses and copy gains were considered hemizygous deletions and low-level gains, respectively. Gene-level CNA of a coding gene was identified by an overlap of at least 50% of the transcript length with a copy number segmentation call. A catalog of gene-level driver homozygous deletions and amplifications detected in the ICGC cohort had been made available in the original study by Nik-Zainal et al. We performed a concordance check by calculation of the Cohen's Kappa

coefficient between this list of oncogenic CNAs and the gene-level CNAs generated using our definition restricted to homozygous deletions and amplifications of the same genes. A Cohen's Kappa coefficient of 0.819 was achieved, indicating an excellent agreement between the two lists of events. We therefore proceeded to use our extended set of gene-level CNAs including all four categories of CNA calls in subsequent analyses. In downstream analyses involving the two cohorts METABRIC and ICGC, we considered homozygous deletions, hemizygous deletions, and amplifications as oncogenic events, while low-level gains were treated equivalently to no change in copy number (neutral copy number). Gene-level CNA events of the ELBC cohort were available for retrieval from the original publication. Events in this dataset were, however, only distinguished between copy gains and copy losses. Hence, we adopted this existing classification and considered both copy gains and copy losses oncogenic events in downstream analyses of CNA data for this particular cohort. Gene-level CNAs included in downstream analyses were limited to those involving genes that were previously reported to harbor driver CNAs in primary BC by Nik-Zainal et al.

### Co-occurrence and mutual exclusivity analyses of somatic genomic alterations

A Poisson–Binomial distribution-based analysis implemented in the R package 'Rediscover' (v0.2.0) was performed to identify co-occurring or mutually exclusive pairs of somatic oncogenic alterations[73]. Owing to the fact that homozygous deletions were very rare, we concentrated on the analyses of gene mutations, gene-level amplifications, and gene-level hemizygous deletions. For each of these three types of alterations and each of the three patient subgroups, a matrix containing expected probabilities per gene per sample was estimated. These subgroup-specific probability matrices and binary matrices indicating the presence or absence of alterations in tumors from patients of each BMI category of the respective patient subgroup were used as the input for pairwise estimation of $p$ values. The corresponding null hypothesis is that the two tested alterations occur independently of each other. Pairs of a gene mutation and a gene mutation, a gene mutation, and a gene-level amplification, a gene mutation and a gene-level hemizygous deletion, in which both alterations occurred at least three times in the respective sub-cohort, were evaluated.

### Differential gene expression and gene set enrichment analyses according to BMI

Analysis of bulk transcriptomic data was performed using the R/Bioconductor package 'limma' (v3.48.3) to identify differentially expressed genes (DEGs) according to BMI[74]. Linear models were adjusted for menopausal status (post- vs pre-menopausal) and tumor grade (G3 vs G1/G2). False discovery rate (FDR) was controlled by p-value adjustment using the Benjamin-Hochberg method. DEGs were determined as those having an absolute log-fold change (logFC) ≥ 0.1, $p$ value < 0.0001, and FDR-adjusted $p$ value ($q$ value) <0.1. Particularly for the NST ER−/HER2− subtype, we reported in the main text DEGs were selected with a less stringent cutoff of 0.5 for $q$ value due to a relatively limited number of obese cases.

To explore the association between BMI and the activity level of biological processes, we performed gene set enrichment analysis using two independent approaches: the supervised population-based Gene Set Enrichment Analysis (GSEA−v4.1.0)[75,76] and the unsupervised single sample-based method Gene Set Variation Analysis (R package 'GSVA'− v1.40.1)[77]. The former method was executed using the complete list of genes pre-ranked by the logFC of the prior differential gene expression analysis. Hallmark gene sets available in the H collection of MSigDB (v7.5.1) were used as references.

### Single-cell gene expression analyses

Analyses of raw gene expression matrices including cell clustering were performed by Bassez, Vos et al. using Seurat v3 R package. Seurat objects containing raw data, cluster assignment, cell type, and cell subtype annotation were retrieved and further analyzed using Seurat (v4.1.1). We considered eight cell types in our analyses: Cancer cells, B cells, T cells, Macrophages/Monocytes, Dendritic cells, Mast cells, Fibroblasts, and Endothelial cells. DGEA was performed for each cell type and subtype using the MAST test with the FindMarkers function in Seurat with a threshold of 0.1 for expression in a minimum fraction of cells in each BMI category. DEGs were selected as those with absolute logFC ≥0.5 and $q$ value < 0.05. GSEA was performed on the GOBP and REACTOME gene sets from MSigDB (v7.5.1).

### Cell-cell communication analyses

We explored the intercellular interactions in the TME at single-cell level with the computational prediction of receptor-ligand interactions between cell types. This was performed using the CellChat toolkit and its accompanying curated interaction database[51]. Cell types that were absent in tumors from one of the BMI categories being compared, i.e., lean and obese, were excluded from the analysis.

### Statistical analyses

Statistical analyses were performed using R version 4.1.1. All statistical tests were two-sided.

The heterogeneity in clinicopathological characteristics between the METABRIC and ICGC cohorts, which would be combined in the analysis of genomic alterations to follow, was assessed using Fisher's exact test (see Supplementary Data 1). We evaluated for each of the data cohorts the association of clinicopathological variables, which include age (>50 vs ≤50), tumor grade (G3 vs G1/G2), tumor size (≥2 cm vs <2 cm), nodal status (positive vs negative), and stage (III/II vs I), with categorical BMI and continuous BMI using Fisher's exact test and Kruskal Wallis test, respectively (see Supplementary Data 2).

Firth's logistic regression models were used for association analyses of recurrent somatic alterations, which were implemented using the R package 'logistf' (v1.24.1). Gene mutations with at least 5 events of occurrence and gene-level CNAs with at least 10 events in each stratified subgroup were evaluated. With clinicopathological variables as independent variables, models were either adjusted for cohort (METABRIC vs ICGC) when testing on the combined cohort, or univariable otherwise. With BMI, models were adjusted for cohort, age, and tumor grade, which were selected based on existing knowledge[78,79], and the results of the aforementioned analysis.

Somatic alterations reported to be associated with either continuous or categorical BMI were explored for potential non-linear association with BMI, at univariable and multivariable level. This was done by fitting generalized additive models, with and without a spline term, for each of the evaluated somatic alterations and comparing these two models. The non-linear model was considered if selected by AIC ($AIC_{non-linear} < AIC_{linear}$), and additionally evident in a likelihood-ratio test ($p$ value < 0.05) which is often expected to be more conservative. In case of non-linearity, the log-odds ratio of the event was fitted against continuous BMI, considering a BMI of 20 as the baseline.

### Reporting summary

Further information on research design is available in the Nature Portfolio Reporting Summary linked to this article.

## Data availability

Data from the ICGC cohort (project BRCA-EU) can be accessed through the ICGC Data Portal [https://dcc.icgc.org/projects/BRCA-EU] and through published data (Nik-Zainal et al. Nature 2016). Data from

METABRIC can be accessed through cBioPortal [https://www.cbioportal.org/study/summary?id=brca_metabric] and through published data (Curtis et al. Nature 2012, Mukherjee et al. NPJ Breast Cancer 2018). Data from ELBC can be accessed through published data (Desmedt et al. JCO 2016) and Gene Expression Omnibus (accession number GSE88770). BMI data for the ICGC, METABRIC, and ELBC cohorts were additionally collected and are accessible via the CodeOcean capsule (see Code availability). Data from MINDACT can be accessed through the EORTC ([https://www.eortc.org/data-sharing/]). The download of the read count data per individual patient from BioKey is publicly available at https://lambrechtslab.sites.vib.be/en/single-cell. Raw sequencing reads for the scRNA-seq experiments have been deposited in the European Genome-phenome Archive (EGA) under study no. EGAS00001004809 (with a summary of the BioKey study and patient characteristics) and with data accession no. EGAD00001006608 (to access the data itself under restricted access). Requests for accessing raw sequencing reads and processed data will be reviewed by the UZLeuven-VIB data access committee. Any data shared will be released via a Data Transfer Agreement that will include the necessary conditions to guarantee the protection of personal data (according to European GDPR law). Source data are provided with this paper.

## Code availability

The R code for data analyses is available in a CodeOcean capsule [https://doi.org/10.24433/CO.8331460.v1]. Results generated from the publicly available data cohorts, namely ICGC, METABRIC, and ELBC, can be fully reproduced within the code capsule. For analyses of data cohorts with restricted access, namely MINDACT and Biokey, complete code is shared but partially not executable due to the unavailability of primary data in the code capsule. Instead, supplementary tables containing secondary data, if applicable, were used for reproducing the displayed figures.

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

## Acknowledgements

The study was financially supported by the Luxembourg Cancer Foundation (grant FC/2018/07), the Consolidator Grant approved by the European Research Council (ERC, FAT-BC 101003153), and the Internal Funds KU Leuven (3M180676). K.V.B. and M.D.S. are funded by the KU Leuven Fund Nadine de Beaufort. F.R. and T.G. are funded by FWO through a research fellowship. G.F. is the recipient of a post-doctoral mandate from the Klinsche Onderzoek en OpleidingsRaad (KOOR) of the University Hospitals Leuven. The METABRIC project was funded by Cancer Research UK, the British Columbia Cancer Foundation, and Canadian Breast Cancer Foundation BC/Yukon. The project also received support from the University of Cambridge, Hutchinson Whampoa, the NIHR Cambridge Biomedical Research Centre, the Cambridge Experimental Cancer Medicine Centre, the Centre for Translational Genomics (CTAG) Vancouver, and the BCCA Breast Cancer Outcomes Unit. The ICGC-BRCA project has been funded through the ICGC Breast Cancer Working Group by the Breast Cancer Somatic Genetics Study (BASIS), a European research project funded by the European Community's Seventh Framework Programme (FP7/2010-2014) under the grant agreement number 242006; the Triple Negative project funded by the Wellcome Trust (grant reference 077012/Z/05/Z) and the HER2+ project funded by Institut National du Cancer (INCa) in France (Grants N° 226-2009, 02-2011, 41-2012, 144-2008, 06-2012). The ICGC Asian Breast Cancer Project was funded through a grant from the Korean Health Technology R&D Project, Ministry of Health & Welfare, Republic of Korea (A111218-SC01). The MINDACT trial has received grants from the European Commission Framework Programme VI (FP6-LSHC-CT-2004-503426), the Breast Cancer Research Foundation, Novartis, F. Hoffman La Roche, Sanofi-Aventis, the National Cancer Institute (NCI), the EBCC-Breast Cancer Working Group (BCWG grant for the MINDACT biobank), the Jacqueline Seroussi Memorial Foundation (2006 JSMF award), Prix Mois du Cancer du Sein (2004 award), Susan G. Komen for the Cure (SG05-0922-02), Fondation Belge Contre le Cancer (SCIE 2005-27), Dutch Cancer Society (KWF), Association Le Cancer du Sein, Parlons-en!, Deutsche Krebshilfe, the Grant Simpson Trust and Cancer Research UK. This trial was also supported by the EORTC Cancer Research Fund. Whole genome analysis was provided in kind by Agendia. The BioKey study was supported by an MSD grant to A.S., by Fonds Nadine De Beauffort to A.S., by a 'Kom op Tegen Kanker' to A.S. and H.W., by the Stichting Tegen Kanker and the Flemish Fund for Scientific Research (FWO; project G0B6120N) Belgium, by Agilent Technologies (Thought Leader award) to D.L. This VIB Grand Challenges project also received support from the Flemish Government under Management Agreement 2017–2021 (VR 2016 2312 doc.1521/4), from the European Union's Horizon 2020 Research and Innovation Programme under grant agreement no. 847912 (RESCUER) and from KU Leuven grant (C14/18/092) Symbiosys3. We are grateful to all women who participated and donated tissue in all studies used in this project and their families; all the investigators, surgeons, pathologists, and research nurses; and finally our close collaborators for their help in the data collection process and the collaboration on the scientific work.

## Author contributions

F.R. and C.De. designed the study. S.A., A.Ba, A.Bo, J.B., A.Br, C.C., F.C., M.D., C.A.D., A.M.G., A.R.G., E.I., J.E., H.K., S.Kn, S.Kr, S.R.L., A.L., J.W.M.M., A.E.M.R., L.M., S.N., S.N-Z., I.N, P.N, M.P., C.Po, K.P., C.Pu, E.Ra, A.R., E.Ru, A.V-S., P.T.S., M.K.S., C.S., P.N.S., K.T.B.T., A.T., S.T., M.V.d.V., S.V.L., L.v.V., G.V., A.V., H.V., A.T.W., H.W., A.S., and D.L. contributed samples and data. H-L.N, F.R. performed data analyses with critical inputs from A.D.G, E.B., and C.De. H-L.N., F.R., and C.De. interpreted the data with substantial contributions from T.G., M.M., M.D.S., E.I., S.N., K.V.B., G.F., A.D.G., D.L., E.B., as well as all other authors. H-L.N., F.R., and C.De. wrote the manuscript. All authors read, revised, and approved the manuscript.

## Ethics declaration

Ethical approval was granted for each of the source studies. Data were acquired for the purpose of this study through publications and open-access data portals for the ICGC, METABRIC, and ELBC cohorts. Ethical compliance for the use of data from the MINDACT and BioKey cohorts was ensured through a Data Transfer Agreement approved by the EORTC and UZ Leuven-VIB, respectively.

## Competing interests

The authors declare no competing interests.

## Additional information

Ha-Linh Nguyen [1], Tatjana Geukens[1], Marion Maetens[1], Samuel Aparicio [2,3], Ayse Bassez[4,5], Ake Borg [6,7,8,9], Jane Brock[10], Annegien Broeks[11], Carlos Caldas [12], Fatima Cardoso[13], Maxim De Schepper [1], Mauro Delorenzi[14,15], Caroline A. Drukker[16], Annuska M. Glas [17], Andrew R. Green [18], Edoardo Isnaldi [1], Jórunn Eyfjörð[19], Hazem Khout[20], Stian Knappskog [21,22], Savitri Krishnamurthy[23], Sunil R. Lakhani [24,25], Anita Langerod[26], John W. M. Martens [27], Amy E. McCart Reed [24], Leigh Murphy[28], Stefan Naulaerts [29], Serena Nik-Zainal [30,31], Ines Nevelsteen[32], Patrick Neven[33], Martine Piccart [34], Coralie Poncet[35], Kevin Punie[36], Colin Purdie [37], Emad A. Rakha[38,39], Andrea Richardson[40], Emiel Rutgers[41], Anne Vincent-Salomon [42], Peter T. Simpson [24], Marjanka K. Schmidt [43], Christos Sotiriou [44], Paul N. Span [45], Kiat Tee Benita Tan[46,47,48], Alastair Thompson[49], Stefania Tommasi [50], Karen Van Baelen[1], Marc Van de Vijver[51], Steven Van Laere[52], Laura van't Veer [53], Giuseppe Viale[54,55], Alain Viari[56], Hanne Vos [57], Anke T. Witteveen[17], Hans Wildiers[36], Giuseppe Floris [58], Abhishek D. Garg [29], Ann Smeets [57], Diether Lambrechts [4,5], Elia Biganzoli [59], François Richard [1,60] & Christine Desmedt [1,60] ✉

[1]Laboratory for Translational Breast Cancer Research, Department of Oncology, KU Leuven Leuven, Belgium. [2]Department of Molecular Oncology, BC Cancer Research Centre, Vancouver, BC, Canada. [3]Department of Pathology and Laboratory Medicine, University of British Columbia, Vancouver, BC, Canada. [4]Laboratory for Translational Genetics, Department of Human Genetics, KU Leuven Leuven, Belgium. [5]VIB Center for Cancer Biology, Leuven, Belgium. [6]Department of Clinical Sciences, Division of Oncology and Pathology, Lund University, Lund, Sweden. [7]Lund University Cancer Center Lund, Lund, Sweden. [8]CREATE Health Strategic Centre for Translational Cancer Research, Lund University, Lund, Sweden. [9]Department of Clinical Sciences, SCIBLU Genomics, Lund University, Lund, Sweden. [10]Department of Pathology, Brigham and Women's Hospital, Boston, MA, USA. [11]Departments of Core Facility, Molecular Pathology and Biobanking, Antoni van Leeuwenhoek, the Netherlands Cancer Institute, Amsterdam, the Netherlands. [12]Cancer Research UK Cambridge Institute and Department of Oncology, Li Ka Shing Centre, University of Cambridge, Cambridge, UK. [13]Breast Unit, Champalimaud Clinical Center/Champalimaud Foundation, Lisbon, Portugal. [14]Department of Oncology, University of Lausanne, Epalinges, Switzerland. [15]SIB Swiss Institute of Bioinformatics, Bioinformatics Core Facility, Lausanne, Switzerland. [16]Department of Surgical Oncology, Antoni van Leeuwenhoek Hospital, Amsterdam, the Netherlands. [17]Agendia, Amsterdam, the Netherlands. [18]Nottingham Breast Cancer Research Centre, School of Medicine, University of Nottingham, Nottingham, UK. [19]BioMedical Center, School of Health Sciences, Faculty of Medicine, University of Iceland, Reykjavík, Iceland. [20]Department of Breast Surgery, Glenfield Hospital, University Hospitals of Leicester NHS Trust, Leicester, UK. [21]Department of Clinical Science, Faculty of Medicine, University of Bergen, Bergen, Norway. [22]Department of Oncology, Haukeland University Hospital, Bergen, Norway. [23]Department of Pathology, The University of Texas MD Anderson Cancer Center, Houston, TX, USA. [24]UQ Centre for Clinical Research, Faculty of Medicine, The University of Queensland, Herston, QLD, Australia. [25]Pathology Queensland, The Royal Brisbane and Women's Hospital, Herston, QLD, Australia. [26]Department of Cancer Genetics, Institute for Cancer Research, Oslo University Hospital, Ullernchausseen Oslo, Norway. [27]Department of Medical Oncology and Cancer Genomics Netherlands, Erasmus MC Cancer Institute, Erasmus University Medical Center, Rotterdam, the Netherlands. [28]University of Manitoba and Cancer Care Manitoba Research Institute, Winnipeg, MB, Canada. [29]Laboratory of Cell Stress & Immunity, Department of Cellular & Molecular Medicine, KU Leuven Leuven, Belgium. [30]Department of Medical Genetics, School of Clinical Medicine, University of Cambridge, Cambridge, UK. [31]MRC Cancer Unit, School of Clinical Medicine, University of Cambridge, Cambridge, UK. [32]Department of Surgical Oncology, University Hospitals Leuven, KU Leuven Leuven, Belgium. [33]Department of Gynecological Oncology, University Hospitals Leuven, Leuven, Belgium. [34]Institut Jules Bordet and Université Libre de Bruxelles, Brussels, Belgium. [35]European Organisation for Research and Treatment of Cancer (EORTC) Headquarters, Brussels, Belgium. [36]Department of General Medical Oncology and Multidisciplinary Breast Unit, Leuven Cancer Institute and University Hospitals Leuven, Leuven, Belgium. [37]Department of Pathology, University of Dundee, NHS Tayside, Dundee, UK. [38]Division of Cancer and Stem Cells, School of Medicine, University of Nottingham, Nottingham, UK. [39]Department of Histopathology, Nottingham University Hospital NHS Trust, City Hospital Campus, Nottingham, UK. [40]Johns Hopkins University School of Medicine, Baltimore, MD, USA. [41]Department of Surgical Oncology, Netherlands Cancer Institute, Amsterdam, the Netherlands. [42]Diagnostic and Theranostic Medicine Division, Institut Curie, PSL Research University, Paris, France. [43]Division of Molecular Pathology, Netherlands Cancer Institute—Antoni van Leeuwenhoek Hospital, Amsterdam, the Netherlands. [44]Institut Jules Bordet, Universite Libre de Bruxelles, Brussels, Belgium. [45]Department of Radiation Oncology, Radboud University Medical Center, Nijmegen, the Netherlands. [46]Department of General Surgery, Sengkang General Hospital, Singapore, Singapore. [47]Department of Breast Surgery, Singapore General Hospital, Singapore, Singapore. [48]Department of Breast Surgery, National Cancer Centre, Singapore, Singapore. [49]Department of Surgery, Dan L Duncan Comprehensive Cancer Center, Baylor College of Medicine, Houston, TX, USA. [50]Molecular Diagnostics and Pharmacogenetics Unit, IRCCS Istituto Tumouri "Giovanni Paolo II", Bari, Italy. [51]Department of Pathology, Amsterdam University Medical Centers, Cancer Center Amsterdam, University of Amsterdam, Amsterdam, the Netherlands. [52]Center for Oncological Research (CORE), Integrated Personalized and Precision Oncology Network (IPPON), University of Antwerp, Antwerp, Belgium. [53]Department of Laboratory Medicine, UCSF Helen Diller Family Comprehensive Cancer Center, San Francisco, CA, USA. [54]Division of Pathology, IEO, European Institute of Oncology IRCCS, Milan, Italy. [55]Department of Oncology and Hemato-Oncology, University of Milan, Milan, Italy. [56]Synergie Lyon Cancer, Plateforme de Bio-informatique 'Gilles Thomas', Lyon, France. [57]Department of Surgical Oncology, University Hospitals Leuven, Leuven, Belgium. [58]Department of Pathology, University Hospitals Leuven, Leuven, Belgium. [59]Unit of Medical Statistics, Biometry and Epidemiology, Department of Biomedical and Clinical Sciences (DIBIC) "L. Sacco" & DSRC, LITA Vialba campus, Università degli Studi di Milano, Milan, Italy. [60]These authors jointly supervised this work: François Richard, Christine Desmedt. ✉e-mail: christine.desmedt@kuleuven.be

