## [Peer review file · Nature Communications]

Obesity-associated changes in molecular biology of primary breast cancerREVIEWER COMMENTS

Reviewer #1 (Remarks to the Author):

This manuscript describes efforts to determine the molecular changes associated with breast tumorigenesis in both the tumor and microenvironment by adiposity/BMI status. The authors used data from 5 previously studied cohorts, including women with breast cancer who were not underweight. I found this to be a timely and interesting manuscript which contained a large amount of data.

The results are dense. I like figure 1, however, it would be helpful to include a table summarizing the results presented in the Association of BMI with driver mutations section.

Could the authors speculate about the handful of alterations that are significantly different between lean and overweight people (e.g. PTEN) but there is no difference (reported) between obese and lean? Similarly, given the similar DEG patterns and TME when looking at lean to overweight compared to lean to obese (line 322-323 and line 417-418), is it the transition to an overweight state that is more important in breast tumor molecular underpinnings rather than reaching the threshold of obesity?

Minor points:

A few language issues (e.g. abstract lines 1-2: don't need "yet" as started with "while"; same thing in lines 139-140; line 131/132 you use both present and past tense in the same sentence). Line 425 "in human" should be in humans

Reviewer #2 (Remarks to the Author):

Nguyen et al completed a retrospective analysis of published and publicly available data on breast cancer patients. This study aims to determine obesity associated molecular changes in breast tissue of these cancer patients. The study identified several BMI associated genomic alterations (somatic mutations, CNAs) in cancer patients. Some of these changes are predicted to be clinical targets.

Authors proposed that BMI influences the global transcriptome of tumour microenvironment of breast cancer patients. Towards this they interrogated single cell RNAseq profiles of biopsies taken from treatment naïve breast cancer patients from a published clinical study. The analysis revealed distinct molecular phenotype of fibroblasts, T cells and macrophages based on BMI and hormone receptor status.

Comments: I request authors to address following queries:

- ScRNAseq data from the cancer cohort was analysed without healthy or non-malignant controls. Inclusion of single cell data from non-malignant (>30 BMI) individuals will be critical to determine clinical relevance of differentially expressed genes and pathways identified in this study.

- Regarding Biokey data- It is interesting that the multidirectional immunoregulatory signalling was observed in cancer patients (with high BMI). Did the authors correlate these observations with the treatment response for these patients?

- Provide details on cell clusters identified in scRNAseq analyses i.e total number of cells detected for each epithelial/cancer, immune and fibroblast cell clusters.

- In the abstract, it is proposed that patient adiposity might play role in regulating heterogeneity of breast cancer. However, it is not clear from the single cell data results whether changes in cellular heterogeneity in cancer cell or immune stromal cell populations was analysed? Integrated scRNAseq UMAP or tSNE analysis of breast cancer samples (lean, overweight and obesity) might reveal changes

in heterogeneity in different cell types present in TME. This information will strengthen this study.

- The molecular changes in TME are partly regulated by factors such as tumour cell diversity. Was there any correlation between increasing BMI and cancer cell heterogeneity in the Biokey data?

- Adipocytes play an important role in determining the adipose tissue function. Absence of adipocytes in single cell profile leaves a gap in the knowledge in terms of fully understanding the extent of BMI driven transcriptomic changes in breast tissue of BC patients.

Reviewer #1 (Remarks to the Author):

This manuscript describes efforts to determine the molecular changes associated with breast tumorigenesis in both the tumor and microenvironment by adiposity/BMI status. The authors used data from 5 previously studied cohorts, including women with breast cancer who were not underweight. I found this to be a timely and interesting manuscript which contained a large amount of data.

The authors would like to thank the reviewer for taking the time to review the manuscript. We acknowledge that all the comments present very valid points and appreciate the invaluable feedback.

Comments:

The results are dense. I like figure 1, however, it would be helpful to include a table summarizing the results presented in the Association of BMI with driver mutations section.

We are aware of the density of the data presented in the manuscript and therefore provide a summary of the highlighted findings in Figure 7 (where the first column is dedicated to the findings on driver mutations). Figure 1 presents genes associated with either continuous or categorical BMI with statistical evidence ($p < 0.05$). Similar data but extended to all breast cancer-specific driver gene mutations are shown in Extended Figure 1, 2, 3, for the NST ER+/HER2-, NST ER-/HER2- and ILC ER+/HER2- subgroups, respectively. Shall there be further improvement that needs to be implemented, we would appreciate additional specifications from the reviewer.

Could the authors speculate about the handful of alterations that are significantly different between lean and overweight people (e.g. PTEN) but there is no difference (reported) between obese and lean?

It is indeed a noteworthy observation that several genomic alterations showed altered frequencies in overweight compared to lean, but not in obese.

We hypothesize that there could be a selection of different oncogenic mutagenesis processes and gene mutations, not only between obese versus lean or overweight versus lean, but also between overweight versus obese. Mechanisms could also vary among patients according to various factors. An instance could be the development of obesity/overweight due to hereditary versus lifestyle factors, which might predispose cells to different modifications to their cellular processes leading to mutagenesis. Germline genomic data would be required to gain more insights in this case. These hypotheses could not be investigated using the data in the current study. Nevertheless, exploratory association studies such as ours are needed for guiding research questions of mechanistic studies. We have additionally discussed the relevance of mechanistic investigation of these obesity/overweight-associated genomic alterations in the revised manuscript (line 445-447: “Furthermore, future studies of mechanisms underlying different selection of oncogenic genomic alterations according to adiposity status are warranted for better comprehension of these associations”).

Additionally, we would like to re-emphasize that measurements beyond BMI need to be explored and compared to BMI in terms their indicative value of patients' adiposity to validate the non-linear/non-monotonic patterns along the BMI spectrum observed in this study.

Similarly, given the similar DEG patterns and TME when looking at lean to overweight compared to lean to obese (line 322-323 and line 417-418), is it the transition to an overweight state that is more important in breast tumor molecular underpinnings rather than reaching the threshold of obesity?

Regarding the similar gene expression profile and TME between overweight and obese as compared to lean, we speculate that it is indeed the case that some molecular features might already undergo biologically meaningful changes transitioning from a lean state to an overweight state, but not all. The threshold could vary between different biological processes depending on their sensitivity to the metabolic environment. In the current manuscript, we aim to give a global view of the association of BMI with tumor molecular biology, which may provide suggestions for further investigations with regard to specific clinical targets where more precise thresholds can be determined. Once more, adiposity measurements other than BMI need to be explored in this regard.

Minor points: A few language issues (e.g. abstract lines 1-2: don't need "yet" as started with "while"; same thing in lines 139-140; line 131/132 you use both present and past tense in the same sentence). Line 425 "in human" should be in humans.

We thank the reviewer for pointing out these issues. They have been fixed in the revised manuscript.

Reviewer #2 (Remarks to the Author):

Nguyen et al completed a retrospective analysis of published and publicly available data on breast cancer patients. This study aims to determine obesity associated molecular changes in breast tissue of these cancer patients. The study identified several BMI associated genomic alterations (somatic mutations, CNAs) in cancer patients. Some of these changes are predicted to be clinical targets.

Authors proposed that BMI influences the global transcriptome of tumour microenvironment of breast cancer patients. Towards this they interrogated single cell RNAseq profiles of biopsies taken from treatment naïve breast cancer patients from a published clinical study. The analysis revealed distinct molecular phenotype of fibroblasts, T cells and macrophages based on BMI and hormone receptor status.

The authors would like to thank the reviewer for taking the time to review the manuscript. We acknowledge that all the comments present very valid points and appreciate the invaluable feedback.

Comments: I request authors to address following queries:

- ScRNAseq data from the cancer cohort was analysed without healthy or non-malignant controls. Inclusion of single cell data from non-malignant (>30 BMI) individuals will be critical to determine clinical relevance of differentially expressed genes and pathways identified in this study.

The reviewer has brought up an important point on the need for coupling the investigation of the impact of BMI on the biology of breast tumors with that of normal breast tissues. Unfortunately, data for healthy samples were not available within the scope of the Biokey study as an ethical approval for pre-treatment collection and analysis of healthy mammary tissues was not granted for this study. We performed a survey of literature and open-source data portals and identified five publicly available scRNA-seq datasets for non-malignant breast tissues¹⁻⁵. The majority of these datasets were not suitable for this purpose due to either unavailable BMI data (three datasets)^{2,3,5}, or inclusion bias (one dataset where samples were obtained from patients undergoing reduction mammoplasty, who are more prone to obesity and younger than the general breast cancer population)⁴.

We then looked into the only relevant study for our needs, the one by Bhat-Nakshatri et al, who analyzed scRNA-seq data generated for normal breast tissues from 12 donors, among whom BMI was available for 11¹. scRNA-seq data of 6 freshly collected samples, whose donors could be individually identified (2 lean, 3 overweight, 1 obese), and another 5 cryopreserved samples, which were pooled and analyzed together, were available. We retrieved the processed data (quality controlled, normalized, clustered and annotated) published in the Human Cell Atlas data portal (<https://data.humancellatlas.org/explore/projects/a004b150-1c36-4af6-9bbd-070c06dbc17d>).

Lymphocyte-related, endothelial and fibroblast-like cells were excluded in the published Seurat object. We plotted the UMAP of the annotated epithelial subtypes, basal/stem, luminal progenitor, and mature luminal, from the six donors with known BMI (Additional Figure 1).

We then also performed differential gene expression analyses (DGEA) for all epithelial cells, and the three epithelial subtypes. The volcano plots in the Additional Figure 2 show the obese vs lean DEGs detected in each of these analyses. The overlaps between obesity-associated DEG profiles in the Biokey cancer cells and normal epithelial cells from Bhat-Nakshatri et al were marginal, although there were indeed several genes that were found to be upregulated by obesity in both neoplastic and normal epithelial cells, e.g. *CXCL13*, *APOD*. Direct comparison of the results derived from these two datasets might be confounded by inter-individual variability and severe batch effects as data were processed using different workflows. We therefore decided against including these sup-optimal analyses in the current manuscript, while taking cautions in the interpretation of current results. In the Results section dedicated to the transcriptomic profile of cancer cell population, we speculated that “these detected changes hint at a possible reprogramming of mammary epithelial cells in an obese setting via a complex and varying combination of cellular and metabolic processes” (line 360-361). We did not attempt to draw definite conclusions regarding specific clinical implications of the observations made by the current analyses, but suggested that “observations implying obesity-driven changes in the expression profile of the cancer cell population, which might lead to changes in the behavior of the disease in biological, prognostic and therapeutic contexts, were made and could be further investigated and validated” (line 473-476), as well as “the nature of these changes was different according to the ER-status based on our current data, which might hypothetically have major repercussions for treatment strategy. Further validation and mechanistic investigation in this direction could pave the way for designing therapeutic combinatorial treatments against BC in the obesity context” (line 480-484). A statement was added to the Discussion section to

explicitly stress the need for analyses of non-malignant controls (line 484-485: “Importantly, the observational findings of this study need to be extended with analyses of healthy controls”).

Fully recognizing the importance of this matter, we are currently actively addressing it in our prospective study FATLAS (<https://clinicaltrials.gov/ct2/show/NCT04200768>). FATLAS was designed to extensively investigate the association between the biology of primary untreated breast cancer and patient’s adiposity. The characterization of the tumor and tumor microenvironment at the single-cell level comprises a major part of this study. In FATLAS, we are systematically collecting tumor tissue and histologically normal tissue adjacent to tumor from each of the patients enrolled. Single-nuclei RNA-seq data for all collected tumor and tumor-adjacent normal samples are being generated. Using tumor-adjacent normal tissue as controls offers a better restriction of inter-individual variability and easier accessibility of samples given less ethical challenges⁶. It also allows discovery of therapeutic targets associated with the interaction between tumor and its adjacent stroma⁶. We additionally mentioned this in the Discussion (line 499-500: “Tumor-adjacent normal tissues will also be available and analyzed within the scope of this study”). The results of the FATLAS study, which is still ongoing and recruiting patients, will be the subject of a separate manuscript.

Additional Figure 1. UMAP of all epithelial cells from donors with known BMI in Bhat-Nakshatri et al color-coded by donor (a), BMI category (b), and epithelial subtype (c).

Additional Figure 2. Differential gene expression of epithelial cell type and subtypes in normal breast tissues from obese versus lean donors in Bhat-Nakshatri et al.

- Regarding Biokey data- It is interesting that the multidirectional immunoregulatory signalling was observed in cancer patients (with high BMI). Did the authors correlate these observations with the treatment response for these patients?

It is indeed one of the ultimate goals to evaluate the translational value of adiposity in terms of treatment response. The Biokey trial was the only study among those in this work for which post-treatment molecular data were available. However, as this was a window-of-opportunity trial where a single dose of the PD1-

blockade pembrolizumab was administered to patients before their surgeries, the actual clinical benefit of the drug could not be assessed using an established clinical endpoint. Bassez et al instead considered T cell clonotype expansion (Expander vs Non-Expander) as a hypothetical surrogate, for which we did not see a noticeable association with BMI in any of the subtypes of interest (Additional Table 1). This lack of correlation might be linked to the multidirectional immunoregulatory signaling observed in the analyses. However, we regrettably would like to save any speculation for the future because firstly, T cell clonotype expansion has not yet been confirmed to be a clinical endpoint, and secondly, the number of patients in this cohort was rather small.

Our goal is to further investigate this in the scope of two neoadjuvant clinical trials where patients are treated with neoadjuvant immunotherapy, pathological complete response considered as clinical endpoint and for which pre and post-treatment samples will be investigated at the single-cell level. As these trials are still running, we expect to investigate this in the coming 1-2 years. Regarding this manuscript, we would like to limit its scope to the characterization of the biology of primary untreated breast cancer.

Additional Table 1. Numbers of patients with (E) and without (NE) T cell clonotype expansion observed during anti-PD1 treatment in the Biokey cohort according to BMI category. scTCR-seq data were available for 21/25 patients whose scRNA-seq data were analyzed in this study.

		lean	overweight	obese
NST ER+/HER2-	E	1	0	1
	NE	5	3	2
NST ER-/HER2-	E	2	1	1
	NE	3	1	1

- Provide details on cell clusters identified in scRNAseq analyses i.e total number of cells detected for each epithelial/cancer, immune and fibroblast cell clusters.

In the current manuscript, the cell counts for all considered cellular compartments, stratified by investigated subtype and BMI category, are provided in Supplementary Table 18. We would appreciate additional specifications from the reviewer if the provided data are not sufficient.

- In the abstract, it is proposed that patient adiposity might play role in regulating heterogeneity of breast cancer. However, it is not clear from the single cell data results whether changes in cellular heterogeneity in cancer cell or immune stromal cell populations was analysed? Integrated scRNAseq UMAP or tSNE analysis of breast cancer samples (lean, overweight and obesity) might reveal changes in heterogeneity in different cell types present in TME. This information will strengthen this study.

In the current manuscript, we aimed to show the heterogeneity in terms of quantitative proportion and expression profile of the cancer and major stromal cell types, i.e. B cells, dendritic cells, fibroblasts, mast cells, monocytes/macrophages, and T cells, according to BMI. The UMAP visualization of all major cell populations from patients analyzed in this manuscript has been added as Supplementary Figure 18. These projections in general demonstrate consistent observations in terms of cellular composition of the TME as those shown in the current main data. However, the variability of total cell counts and counts in each cell population between patients could not be easily taken into consideration, and the differences in the transcriptomic profile could not be deduced from this 2D projection. Therefore, we believe that the results would be better represented by the volcano plots and relative frequency boxplots in the current main figures (Figure 5a-o, Figure 6a, e).

We presumed that the reviewer would like to additionally see the obesity-associated differences in the composition and transcriptomic profile of the cell subtypes within each of the major cell types. This would indeed provide significantly more comprehensive insights into the TME of breast tumors. Subtypes of immune stromal cells, in particular T cells, dendritic cells, and monocytes/macrophages, were identified in re-clustering analyses and annotated according to their markers in the Biokey study (respective UMAPs are displayed in Figure 1e, 5a, 5h in Bassez et al). The cell counts of the annotated cell subtypes are shown in Additional Table 2. Provided the limited number of patients in this series, further breaking down the major cell types into subtypes left several subtypes with low and very low absolute cell counts which might be insufficient for well-powered analyses. It would also not be certain whether the depleted presence or absence of the relatively rare cell subtypes in a patient was due to a biologically meaningful association with BMI or due to technical dropouts. We reasoned that by including more patients and generating single-cell data with higher target cell recovery (more cells loaded per sample), we would be able to systematically and exhaustively examine stromal cell sub-populations with respect to adiposity in FATLAS. This has been discussed in the manuscript as one of the future directions for extending the current findings (line 497-499: “In-depth characterization of all cell populations present in the BC tissue, including adipocytes, more scarce immune cell types such as mast cells and dendritic cells, as well as their phenotypes, will be performed in this study”). On the other hand, precautions would need to be taken for the current data. Regardless, we have been able to present a significant amount of insights into the transcriptomic profile and dynamics of the TME according to BMI at the major cell type resolution in this work, and therefore would like to reassert its relevance and originality.

Additional Table 2. Cell counts of subtypes of dendritic cells, monocytes/macrophages, and T cells in NST ER+/HER2- and NST ER-/HER2- tumors according to BMI category in the Biokey cohort

Cell type	Cell subtype	NST ER+/HER2-			NST ER-/HER2-		
		lean	overweight	obese	lean	overweight	obese
Dendritic cell	AS_DC	4	4	1	11	3	2
	cDC1	14	26	16	46	16	9
	cDC2	121	207	122	445	127	130

	LanghDC	0	4	2	32	47	2
	mregDC_migDC	4	10	12	29	3	11
	plasmacytoid_DC	20	41	38	93	29	42
	quiescent-migDC	147	141	103	156	45	21
Mono/Mf	Mf_CCL18	232	236	117	322	65	92
	Mf_CCL2	1	1	8	71	28	71
	Mf_CCR2	34	29	13	159	44	147
	Mf_CX3CR1	108	90	93	228	38	78
	Mf_LYVE1	22	53	33	75	18	10
	Mf_MMP9	15	26	15	70	4	80
	Mf_MT1G	3	1	2	115	6	29
	Mf_SLC2A1	0	1	2	184	40	42
	Mono_CD14	70	30	32	113	16	38
	Mono_CD16	6	2	6	11	4	4
T cell	CD4_EM	766	927	989	1551	624	317
	CD4_EX	220	286	369	423	275	171
	CD4_N	355	469	525	649	220	143
	CD4_REG	296	494	390	1037	159	336
	CD8_EM	1077	953	1062	1497	438	393
	CD8_EMRA	10	121	62	46	15	22
	CD8_EX	68	66	156	975	41	485
	CD8_N	31	65	29	62	15	4
	CD8_RM	446	369	232	498	189	236
	gdT	59	332	104	148	104	27
	NK_CYTO	52	37	45	47	16	9
	NK_REST	275	155	117	208	79	89
	Proliferating	14	60	23	86	18	78
	Vg9Vd2_gdT	175	79	79	80	69	32

The molecular changes in TME are partly regulated by factors such as tumour cell diversity. Was there any correlation between increasing BMI and cancer cell heterogeneity in the Biokey data?

We pooled all cancer cells identified in the Biokey dataset together for a re-clustering. The following potential confounding factors were regressed out prior to dimensionality reduction and re-clustering: UMI count, proportion of mitochondrial genes, cell cycle scores (S score and G2M score determined by the Seurat CellCycleScoring function), and patient. The UMAPs of cells from patients with NST ER+/HER2- and NST ER-/HER2- are shown in Additional Figure 3. We noted that although the data had been regressed for individual patient, majority of the clusters were still strongly individual-specific. Similar observations were made by Bassez et al (Extended Data Figure 1) and by Wu et al (Figure 1)^{7,8}. Furthermore, Wu et al, concluded that unsupervised clustering, as done here, would not be able to rule out the inter-patient heterogeneity in order to identify intra-tumoral cancer cell subpopulations of the same underlying

molecular profiles across all tumors. This in most cases requires a supervised approach using pre-defined scRNAseq-compatible molecular signatures for sub-populations of interest, for example, intrinsic subtyping signatures in Wu et al. Single-cell DNA sequencing data could also provide information for identification of intra-tumoral subclones, but was not available for this cohort. Provided the complexity of this subject, we would like to dedicate another study for its investigation.

Additional Figure 3. UMAP of cancer cells in tumors from NST ER+/HER2- and NST ER-/HER2- patients from the Biokey cohort. (a) UMAP, computed from data that were not regressed for patient, of all cancer cells color-coded according to patient. (b-e) UMAP, computed from data that were regressed for patient, of all cancer cells color-coded according to patient (b), BMI category (c), tumor subtype (d), and derived clusters (e).

- Adipocytes play an important role in determining the adipose tissue function. Absence of adipocytes in single cell profile leaves a gap in the knowledge in terms of fully understanding the extent of BMI driven transcriptomic changes in breast tissue of BC patients.

We fully acknowledge the fact that characterization of adipocytes and their roles in the TME in a single cell setting is one of the crucial elements in understanding the adiposity-associated BC biology as this is the most abundant cell type in the breast tissue. It is however challenging to capture mature adipocytes from human samples in a regular scRNA-seq workflow as they tend to be lost during cell dissociation given their large sizes, high lipid content, and fragile membranes^{9,10}. Fortunately this technical limitation can possibly be resolved by using several workarounds, one of which is the single-nuclei RNA-seq (snRNA-seq) technique^{10,11}. In FATLAS, snRNA-seq is being employed for data generation, so that adipocytes in both tumor tissues and adjacent normal tissues will be characterized in the function of adiposity measurements.

Concluding remarks from the authors: We completely agree with all the points raised by the reviewers, especially those regarding the scRNA-seq analyses. Due to the retrospective setup of this study, we had to reserve some of the additions suggested for future studies. We fully acknowledged the current limitations of this study, therefore we actively discussed them in the manuscript and took them into consideration in the data interpretation, as well as suggested future directions to address them. While doing so, we insist on the importance of this study as it revealed a significant number of findings derived from analyses of patient data at genomic and transcriptomic levels. Especially, it demonstrated the remarkable potential of the single-cell approach for uncovering clinically meaningful insights into the obesity-breast cancer interplay, which is one of the main take-home messages of the manuscript. Although further investigations are warranted for the current findings before they can be ultimately translated to the improvement of patient care, there is first a need for accumulation of such results in literature. We hope that this manuscript will further contribute to advocating for more research in this topic, so that its data repertoire and knowledgebase will be expanded.

References

1. Bhat-Nakshatri, P. *et al.* A single-cell atlas of the healthy breast tissues reveals clinically relevant clusters of breast epithelial cells. *Cell Rep Med* **2**, 100219 (2021).
2. Nguyen, Q. H. *et al.* Profiling human breast epithelial cells using single cell RNA sequencing identifies cell diversity. *Nat Commun* **9**, 2028 (2018).
3. Pal, B. *et al.* A single-cell RNA expression atlas of normal, preneoplastic and tumorigenic states in the human breast. *EMBO J* **40**, e107333 (2021).
4. Murrow, L. M. *et al.* Mapping hormone-regulated cell-cell interaction networks in the human breast at single-cell resolution. *Cell Syst* **13**, 644-664.e8 (2022).
5. Geldhof, V. *et al.* Single cell atlas identifies lipid-processing and immunomodulatory endothelial cells in healthy and malignant breast. *Nat Commun* **13**, 5511 (2022).
6. Aran, D. *et al.* Comprehensive analysis of normal adjacent to tumor transcriptomes. *Nat Commun* **8**, 1077 (2017).
7. Bassez, A. *et al.* A single-cell map of intratumoral changes during anti-PD1 treatment of patients with breast cancer. *Nat Med* **27**, 820–832 (2021).
8. Wu, S. Z. *et al.* A single-cell and spatially resolved atlas of human breast cancers. *Nat Genet* **53**, 1334–1347 (2021).
9. Deutsch, A., Feng, D., Pessin, J. E. & Shinoda, K. The Impact of Single-Cell Genomics on Adipose Tissue Research. *Int J Mol Sci* **21**, 4773 (2020).
10. Wang, T., Sharma, A. K. & Wolfrum, C. Novel insights into adipose tissue heterogeneity. *Rev Endocr Metab Disord* **23**, 5–12 (2022).

11. Sun, W. *et al.* snRNA-seq reveals a subpopulation of adipocytes that regulates thermogenesis. *Nature* **587**, 98–102 (2020).

REVIEWERS' COMMENTS

Reviewer #1 (Remarks to the Author):

Comments addressed.

Reviewer #2 (Remarks to the Author):

- The authors have justified the non-inclusion of a healthy dataset in these analyses. One of the main reasons is the absence of a publicly available large scRNAseq dataset with BMI information. I thank the authors for making appropriate changes in the result and discussion sections to reflect the limitations of this study.

- In additional Figure 1, the BMI category seems to affect the proportion of basal and luminal cells (middle panel). It looks like lean samples are enriched with luminal cells, including progenitors. However, I can understand that a small sample size limits these analyses.

- I request authors break down the count numbers for individual patient samples in Supplementary Table 18. This is to rule out that one or two samples do not influence the frequency differences shown in Figures 6a and e.

- Figure 7 is wordy with small font size. I will suggest simplifying it for easy reading

Reviewer #1 (Remarks to the Author):

Comments addressed.

We sincerely thank the reviewer for their time taken to re-evaluate the revised manuscript. We are glad that we were able to address their comments and make the necessary changes to improve the manuscript.

Reviewer #2 (Remarks to the Author):

We sincerely thank the reviewer for their time taken to re-evaluate the revised manuscript. We are glad that we were able to make the necessary changes based on their constructive comments to improve the manuscript.

- The authors have justified the non-inclusion of a healthy dataset in these analyses. One of the main reasons is the absence of a publicly available large scRNAseq dataset with BMI information. I thank the authors for making appropriate changes in the result and discussion sections to reflect the limitations of this study.

- In additional Figure 1, the BMI category seems to affect the proportion of basal and luminal cells (middle panel). It looks like lean samples are enriched with luminal cells, including progenitors. However, I can understand that a small sample size limits these analyses.

We thank the reviewer for pointing out these observations, however, it would be indeed difficult to draw any definitive conclusion given the very small sample size of this particular dataset. We also appreciate their objective understanding of the current limitations of the study, which we will definitely address in our follow-up study.

- I request authors break down the count numbers for individual patient samples in Supplementary Table 18. This is to rule out that one or two samples do not influence the frequency differences shown in Figures 6a and e.

We have added a complete table containing the cell counts of considered cell types recovered from each individual tumor in the Biokey cohort as Supplementary Table 19. In each of the boxplots in Figures 6a and e, the cell frequencies shown were computed on a single-tumor basis and are independent of each other. Particularly, each data point represents the frequency of the corresponding cell type detected in an individual sample. In this manner, variability in the cell recovery rate (total number of cells isolated from a biopsy) among samples, which is common in single-cell experiments, can be accounted for. This has also been additionally specified in the legends of Figure 6 and Supplementary Table 19.

- Figure 7 is wordy with small font size. I will suggest simplifying it for easy reading.

We appreciate the suggestion and have reduced the text as well as increase the font size for better readability.